

# Chemical characterisation of benzene oxidation products under high and low NOx conditions using chemical ionisation mass spectrometry

Michael Priestley[1*], Thomas J. Bannan[1], Michael Le Breton[1†], Stephen D. Worrall[1=], Sungah Kang[2], Iida

Pullinen[2+], Sebastian Schmitt[2], Ralf Tillmann[2], Einhard Kleist[7], Defeng Zhao[2-], Jürgen Wildt[2,7], Olga Garmash[4], Archit Mehra[1^], Asan Bacak[1~], Dudley E. Shallcross[3,5], Åsa Halquist[8], Mikael Ehn[4], Astrid Kiendler-Scharr[2], Thomas F. Mentel[2], Gordon McFiggans[1], Mattias Halquist[6], Hugh Coe[1], Carl J. Percival[1l]

[1]Centre for Atmospheric Science, Department of Earth and Environmental Sciences, University of Manchester,
Manchester, M13 9PL, UK

[2]Institut für Energie und Klimaforschung, IEK-8, Forschungszentrum Jülich, Jülich, Germany

[3]School of Chemistry, The University of Bristol, Cantock's Close BS8 1TS, UK

[4]Institute for Atmospheric and Earth System Research/Physics, Faculty of Science, University of Helsinki, 00014 Helsinki, Finland

[5]Department of Chemistry, University of the Western Cape, Bellville, South Africa

[6]Department of Chemistry and Molecular Biology, University of Gothenburg, 412 96, Gothenburg, Sweden (* Now at)

[7]Institut für Bio- und Geowissenschaften, IBG-2: Pflanzenwissenschaften,Forschungszentrum Jülich GmbH, Jülich, Germany

[8]IVL Swedish Environmental Research Institute, Gothenburg, Sweden

[l] Now at Jet Propulsion Laboratory, 4800 Oak Grove Drive, Pasadena, CA 91109

[+] Now at Department of Applied Physics, University of Eastern Finland, 702132 Kuopio, Finland

[=] Now at Aston Institute of Materials Research, School of Engineering and Applied Science, Aston University,
Birmingham, B4 7ET, UK

[†] Now at Volvo Group Trucks Technology, L3 Lundby, Gothenburg, Sweden

[~] Now at Turkish Accelerator & Radiation Laboratory, Ankara University, Institute of Accelerator, Technologies Gölbaşı Campus, 06830 Golbasi / Ankara, Turkey

[^] Now at Faculty of Science and Engineering, University of Chester, CH2 4NU, UK

[-] Now at Department of Atmospheric and Oceanic Sciences & Institute of Atmospheric Sciences, Fudan University, Shanghai, China

Corresponding author: Michael Priestley (michael.priestley@gu.se)

## Abstract

Aromatic hydrocarbons are a class of volatile organic compounds associated with anthropogenic activity and make up a significant fraction of urban VOC emissions that contribute to the formation of secondary organic aerosol (SOA). Benzene is one of the most abundant species emitted from vehicles, biomass burning and industry. An iodide time of flight chemical ionisation mass spectrometer (ToF-CIMS) and nitrate ToF-CIMS were deployed





at the Jülich plant chamber as part of a series of experiments examining benzene oxidation by OH under high and low $NO_x$ conditions, where a range of organic oxidation products were detected. The nitrate scheme detects many oxidation products with high masses ranging from intermediate volatile organic compounds (IVOC) to extremely low volatile organic compounds (ELVOC), including $C_{12}$ dimers. In comparison, very few species with $C_{\geq 6}$ and $O_{\geq 8}$ were detected with the iodide scheme, which detected many more IVOC and semi volatile organic compounds (SVOC) but very few ELVOC and low volatile organic compounds (LVOC). 132 and 195 CHO and CHON oxidation products are detected by the iodide ToF-CIMS in the low and high $NO_x$ experiments respectively. Ring breaking products make up the dominant fraction of detected signal (89 – 91%). 21 and 26 of the products listed in the master chemical mechanism (MCM) were detected and account for 6.4 – 7.3 % of total signal. The time series of highly oxidised ($O_{\geq 6}$) and ring retaining oxidation products ($C_6$ and double bond equivalent = 4) equilibrate quickly characterised by a square form profile, compared to MCM and ring breaking products which increase throughout oxidation exhibiting saw tooth profiles. Under low $NO_x$ conditions, all CHO formulae attributed to radical termination reactions of $1^{st}$ generation benzene products and $1^{st}$ generation autoxidation products are observed, and one exclusively $2^{nd}$ generation autoxidation product is also measured ($C_6H_8O_8$). Several N containing species that are either $1^{st}$ generation benzene products or $1^{st}$ generation autoxidation products are also observed under high $NO_x$ conditions. Hierarchical cluster analysis finds four cluster of which two describe photo-oxidation. Cluster 2 shows a negative dependency on the $NO_2/NO_x$ ratio indicating it is sensitive to NO concentration thus likely to contain NO addition products and alkoxy derived termination products. This cluster has the highest average carbon oxidation state ($\overline{OS_C}$) and the lowest average carbon number and where nitrogen is present in cluster member, the oxygen number is even, as expected for alkoxy derived products. In contrast, cluster 1 shows no dependency on the $NO_2/NO_x$ ratio and so is likely to contain more $NO_2$ addition and peroxy derived termination products. This cluster contains less fragmented species, as the average carbon number is higher and $\overline{OS_C}$ lower than cluster 2, and more species with an odd number of oxygen atoms. This suggests clustering of time series which have features pertaining to distinct chemical regimes e.g. $NO_2/NO_x$ perturbations, coupled with *a priori* knowledge, can provide insight into identification of potential functionality.

## 1.    Introduction

Benzene is an aromatic volatile organic compound (VOC) commonly used as a vehicular fuel additive (Verma and Des Tombe, 2002) and as a chemical intermediate in the manufacture of a range of products e.g. detergents (Oyoroko and Ogamba, 2017), lubricants (Rodriguez *et al.*, 2018), dyes (Guo *et al.*, 2018) and pesticides (Wang *et al.*, 2014). Whilst current global estimates of benzenoid (molecules containing a benzene ring) emission to the atmosphere by vegetation is of a comparable order to that of anthropogenic activity (~ 5 times lower, Misztal et al. 2015), and background concentrations of benzene are enhanced by increased biomass burning (Archibald *et al.*, 2015), emission of benzene to the urban atmosphere is dominated by vehicle exhausts (Gentner *et al.*, 2012) and solvent evaporation (Hoyt and Raun, 2015). As well as a toxin (Snyder, Kocsis and Drew, 1975) and carcinogen (Ruchirawat *et al.*, 2005, 2007; Bird *et al.*, 2010), benzene is photochemically active and contributes to the formation of ozone ($O_3$) and secondary organic aerosol (SOA), both of which act to modify the climate and contribute to poor air quality (Henze *et al.*, 2007; Ng *et al.*, 2007). SOA formation from benzene has previously been quantified with a focus on the contribution from smaller mass, ring breaking reaction products such as





epoxides (Glowacki, Wang and Pilling, 2009) and dicarbonyl aldehydes (Johnson *et al.*, 2005). For example the 37% of the glyoxal formed in the LA basin is estimated to derive from aromatic sources (Knote *et al.*, 2014). These smaller ring breaking products typically make up a larger fraction of SOA mass than the ring retaining

products (Borrás and Tortajada-Genaro, 2012) and so have traditionally been the main focus for SOA quantification. Other major, toxic benzene oxidation products such as catechol, nitrophenol and maleic anhydride are also known components of SOA (Borrás and Tortajada-Genaro, 2012).

More recently the autoxidation mechanism has been demonstrated to occur in aromatic systems (Wang *et al.*, 2017, Wang *et al.*, 2020; Molteni *et al.*, 2018; Tsiligiannis *et al.*, 2019; Garmash *et al.*, 2020; Mehra *et al.*, 2020)

producing highly oxygenated organic molecules (HOM, defined as containing 6 or more oxygen atoms which is a product of the autoxidation mechanism) incorporating up to 11 oxygen atoms (Bianchi *et al.*, 2019). The autoxidation mechanism of intra molecular H shifts from the carbon backbone to the peroxy radical centre forming peroxide groups is consistent with other VOC systems (Rissanen *et al.*, 2014) initiated by a variety of different oxidants (e.g. Mentel et al. 2015; Berndt et al. 2016). The inclusion of an alkyl group to a benzene ring facilitates

HOM formation (Wang *et al.*, 2017) as a consequence of the greater degrees of freedom afforded to the molecule. Whilst benzene is not the most reactive of aromatic VOCs, $\tau_{OH} \approx 9.5$ days for benzene vs. $\tau_{OH} \approx 6\text{-}10$ hours for xylenes ($[OH] = 2.0 \times 10^6$ molecules cm$^{-3}$, Atkinson and Arey, (2007)), it is ubiquitous in the urban environment and is the simplest $C_6$ aromatic ring system to study.

Oxidation of benzene occurs nearly exclusively via hydroxyl radical (OH) addition to form the cyclohexadienyl

radical/benzene-OH adduct, which subsequently adds $O_2$ to form the hydroxycyclohexadiene peroxy radical ($C_6H_6$-OH-$O_2$) (Volkamer *et al.*, 2002) (Fig. 1). Two subsequent reaction pathways are postulated for this peroxy radical: either elimination of $HO_2$ yields phenol and secondary OH attack must occur again; or an endocyclic di-oxygen bridged carbon centred radical intermediate is formed by the addition of the peroxy group to (typically a β-carbon (Glowacki, Wang and Pilling, 2009)). This di-oxygen bridge carbon centred radical may add another $O_2$

and form a peroxy radical (named as BZBIPERO2 in the master chemical mechanism, MCM, Saunders et al. (2003)). However it is not known if autoxidation may continue to form a second oxygen bridged radical, described as type II autoxidation (Molteni *et al.*, 2018, Fig 1, BZBIPERO2-diB), or form a hydroperoxide carbon centred radical, formed through intra molecular hydrogen abstraction (type I autoxidation). At each step, termination of the peroxy radical to hydroxyl, hydroperoxyl, nitrate, peroxy nitrate or reduction to an alkoxy radical is possible,

from which further termination via formation of a nitrite, nitro- or acyl group can occur.

Typically the presence of $NO_x$ alters the product distribution of VOC oxidation and reduces SOA formation (e.g. Stirnweis et al. 2017). With other atmospherically relevant VOC precursors of SOA e.g. isoprene or monoterpenes, high NO conditions can supress SOA formation (Wildt *et al.*, 2014; Sarrafzadeh *et al.*, 2016) as reduction of $RO_2$ to RO ultimately leads to fragmentation of the RO species (Surratt *et al.*, 2006; Nguyen *et al.*,

2015), but it can also form epoxides, aldehydes and hydroperoxides which readily partition to the aerosol phase and contribute to SOA growth (Surratt *et al.*, 2010).

The further reaction of $NO_x$ with oxygenated VOCs produces species including nitro organics, nitrates, peroxy nitrates and peroxyacyl nitrates (Atkinson, 2000) that can also condense and contribute to SOA formation. Under conditions where $NO_x$ is present in moderate amounts, the $HO_2$:OH ratio is low as $HO_2$ is reduced by NO to form

OH. This allows more OH oxidation to occur and less termination of $RO_2$ by $HO_2$ to occur; whereas under low





$NO_x$ conditions VOC consumption is lower, as OH recycling from $HO_2$ relies on the slower $HO_2 + O_3$ reaction (Atkinson, 2000). Here low and high $NO_x$ are relative terms and are defined by the available VOC. This reduction of the $HO_2$:OH ratio as a function of NO has been observed at Mace Head, Ireland where clean low $NO_x$ marine influenced air can be contrasted with polluted $NO_x$ containing continental air (Creasey, Heard and Lee, 2002). In

that instance, an increase in NO concentration from 0.01 ppb to 1 ppb reduces the $HO_2$:OH ratio from 200:1 to 10:1.

The autoxidation mechanism is known to form biogenic HOM, and as a consequence SOA, in ambient rural environments (e.g. Ehn et al. 2014). The mechanism is increasingly a competitive process in urban and suburban environments where $NO_x$ concentrations have seen significant reductions in recent years (Praske *et al.*, 2018).

These low concentrations of $NO_x$ (0.1 - 1's ppb) have been shown to enhance local oxidant production and drive an increase in biogenic HOM formation (Pye *et al.*, 2019). Aromatic HOM formation by OH at relevant urban $NO_x$ concentrations has been demonstrated (Tsiligiannis *et al.*, 2019; Zaytsev *et al.*, 2019; Garmash *et al.*, 2020; Mehra *et al.*, 2020).

ToF-CIMS is a measurement technique frequently used to probe VOC oxidation due to the ability to detect many

low concentration compounds simultaneously in real time (e.g. Chhabra et al. 2015). As a result of the sensitivity of the nitrate ionisation scheme towards HOM, this reagent ion is typically used to study the autoxidation mechanism and HOM formation. However, to achieve carbon mass closure of the system, multiple ionisation schemes are required due to their complimentary yet differing sensitivities towards OVOCs with different oxidation states and functional groups (e.g. Isaacman-VanWertz *et al.*, 2017; Riva *et al.*, 2019).

In this study, the oxidation of benzene by OH under atmospherically relevant high and low urban $NO_x$ conditions is investigated in the Jülich plant atmosphere chamber (JPAC) with two time of flight chemical ionisation mass spectrometers (ToF-CIMS) using the iodide and nitrate ionisation schemes. The properties of the oxidation products are compared between the two ionisation schemes, as well as the high and low $NO_x$ conditions.

## 2. Methods and Experimental

The experiments were performed in a 1450 L borosilicate continuous flow reactor of the Jülich Plant Atmospheric Chamber (JPAC) described elsewhere (Mentel *et al.*, 2009). The chamber is operated in continuous flow mode in order to homogenise the mixture giving an average residence time of ~45 minutes. Benzene (Merck, ≥ 99.7%) was introduced into the chamber by flowing purified ambient air over a diffusion source to maintain a constant concentration and was monitored by recording the inlet and outlet concentrations by quadrupole proton transfer

mass spectrometry (PTR-MS). The PTR-MS (IONICON) was calibrated using a benzene diffusion source (Garmash *et al.*, 2020). The difference between the outlet and inlet concentrations, measured at a 2 minute time resolution and switched every 25 minutes, describes the reacted benzene, from which the OH concentration is calculated.

| Exp | $NO_x$ / ppb | Reacted Benzene / ppb | $J_{(O1D)}$ / $10^{-3}$ $s^{-1}$ | $J_{NO2}$ / $10^{-3}$ $s^{-1}$ | OH / $10^7$ molecules $cm^{-3}$ |
|---|---|---|---|---|---|
| 0 (Low $NO_x$) | ≤ 0.30 | 65.0 | 2.60 | 0.00 | 12.0 |





| 1 (High NOx) | 20.0 | 65.0 | 2.60 | 0.00 | 5.80 |
|---|---|---|---|---|---|
| | | 65.0 | 2.60 | 4.30 | 6.50 |

Table 1. Summary of selected experiments performed at JPAC used in this study. Two OH concentrations are reported for the NOx containing studies. The first describes steady state concentrations with $J_{NO2}$ (UVA) and ultra violet (TUV) light on, the second with only the $J_{NO2}$ (UVA) light on.

OH is generated by photolysis of $O_3$ by a UV lamp ($\lambda < 254$ nm, Philips, TUV 40W herein termed TUV) and the subsequent $O(^1D)$ reaction with $H_2O$. $O_3$ is introduced to the chamber by a separate line to the benzene in order to prevent reactions occurring inline and was monitored by UV absorption (Model 49, Thermo Environmental instruments). OH concentrations are calculated under steady state conditions (ss) according to the equation below and are described in detail elsewhere (Garmash *et al.*, 2020). The residence time in the chamber (t) was 2900 seconds and is equal to the chamber volume (V =1450 L) divided by the flow rate (F) of 30 standard litres per minute (slm) and $k_{Benzene+OH} = 1.23 \times 10^{-12}$ $cm^3$ $s^{-1}$.

$$[OH]_{ss} = \frac{[Benzene]_{in} - [Benzene]_{out}}{[Benzene]_{in}} \frac{1}{k_{Benzene+OH} \left(\frac{V}{F}\right)}$$

$NO_x$ was measured by chemiluminescence (CLD 770 and PLC 760, Eco Physics). For the high $NO_x$ experiment, NO (Linde, 99.5±5 ppm NO in $N_2$) was injected into the chamber via a separate perfluoro alkoxy alkane (PFA) line. Twelve discharge lamps termed UVA (HQI, 400 W/D; Osram, Munich, Germany) were activated to photolyse $NO_2$ ($J_{NO2} = 4.3 \times 10^{-3}$ $s^{-1}$) to increase the $NO/NO_2$ ratio, although $NO_2$ remains the dominant $NO_x$ compound at > 70% of total $NO_x$. The TUV lamp was then switched on and then the UVA lamps switched off. Temperature and humidity were maintained at 288±1 K and 67±2 % through the experiments. The experimental conditions are summarised in Table 1.

It is expected that as the $NO_x$ concentration decreases, peroxy radical lifetime is much longer than the required time scale for autoxidation hydrogen shifts (Praske *et al.*, 2018) and so autoxidation products can be expected in this low $NO_x$ case here (≤ 300 pptv). Where the UVA lamps are active, the $NO/NO_x$ ratio is increased, thus increasing the ratio of $RO/RO_x$ ($RO + RO_2$).

## 2.1.   ToF-CIMS Instrumentation

### Iodide

The University of Manchester ToF-CIMS has been described by Priestley et al. (2018). The iodide ToF-CIMS ion molecule reaction region (IMR) was held at a constant pressure of 100 mbar by a scroll pump (Agilent SH-112) controlled with a servo control valve placed between the scroll pump and the IMR. The short segmented quadrupole (SSQ) region was held at 1.30 mbar by a second scroll pump (Triscroll 600). Chamber air was drawn into the IMR via a critical orifice at 2.2 slm where it mixed with $I^-$ ions. The reagent $I^-$ ions were created by flowing 10 standard cubic centimetres per minute (sccm) of nitrogen ($N_2$) over a methyl iodide ($CH_3I$) permeation tube made of 1/8" PFA and held at 40°C. This flow met 2 slm $N_2$ and was flowed through a $^{210}Po$ 10 mCi alpha emitting reactive ion source (NRD Inc.). The mass calibration was performed for 7 known masses: $NO_3^-$, $I^-$, $I^-.H_2O$, $I^-.HCOOH$, $I^-.HNO_3$, $I_2^-$ and $I_3^-$, covering a mass range of 62 to 381 *m/z*. The mass calibration was fitted to a 3rd order polynomial and was accurate to within 2 ppm, ensuring peak identification was accurate below 20 ppm. The



resolution was 3231 at 127 $m/z$ and 3720 at 381 $m/z$. The inlet to the I⁻ ToF-CIMS sampled from the middle of the chamber and was comprised of 50 cm 1/4" O.D. PFA tubing giving residency time of <1.5 s.

Formic acid calibrations and chamber and instrumental backgrounds were taken before every experiment to assess instrument sensitivity changes. Backgrounds were performed by overflowing the CIMS inlet with $N_2$ and calibrations were performed by flowing 50 sccm $N_2$ over a formic acid permeation tube held at 40°C. The formic acid sensitivity was $3.15 \pm 0.26$ Hz ppt⁻¹ ($2\sigma$) measured as a 5 minute average normalised to $10^6$ Hz iodide. The background was $2.72 \pm 0.66$ ppt ($2\sigma$) measured as a 5 minute average normalised to $10^6$ Hz iodide. The limit of

detection ($3\sigma$) was 100 ppt. Calibration of all detected masses was not possible, so data are presented either as signal (Hz) from the instrument or as a number of species.

It is not possible to identify a compound from its formula alone. Oxidation is initiated with OH addition to the benzene ring, but fragmentation occurs and so the OH moiety may be lost; additionally multiple OH attacks make the counting of oxygen more difficult (Garmash *et al.*, 2020). It is for these reasons that $RNO_x$ species are

described more generally and any discussion of compound classes is speculative. Data analysis was performed using the Tofware (V 3.1.0, (Stark *et al.*, 2015) ).

**Nitrate**

The Jülich ToF-CIMS with nitrate ionisation is described elsewhere (Garmash *et al.*, 2020). Chamber air was drawn into an Eisele type inlet at 9 slm where it merges with a 20 slm sheath flow of nitrogen that contains nitrate

ions ($NO_3$) for approximately 200 ms. These ions are generated by flowing 10 sccm dry nitrogen through a wash bottle containing nitric acid ($HNO_3$, Aldrich, 70%) that pass through a $^{241}$Am alpha emitting source. The inlet flow enters the instrument through a 300 µm diameter aperture after which the ions are collimated and energetically homogenised through a series of differentially pumped chambers.

Mass calibration is performed on nitrate, its dimer and trimer ions, $NO_3$, $NO_3.HNO_3$ and $NO_3.(HNO_3)_2$ and

hexafluoropentanoic acid ($NO_3.HFPA$ adduct) providing an accurate mass range up to 326 $m/z$. The resolution was ~4000 at 125 $m/z$. The inlet to the instrument sampled from the middle of the chamber by a 1m 1/4" O.D stainless steel tube.

### 2.2.    Calculation of average carbon oxidation state ($\overline{OS_c}$)

Average carbon oxidation state $\overline{OS_c}$ (Kroll *et al.*, 2011) is commonly used to describe the degree of oxidation

within a complex oxidation reaction. It is calculated using the oxygen to carbon ratio (O:C), hydrogen to carbon ratio (H:C) and the nitrogen to carbon ratio (N:C) and assumes nitrogen is in the form of nitrate where the oxidation state of N ($OS_N$) is +5 :

$$\overline{OS_c} = (2 \times O{:}C) - H{:}C - (5 \times N{:}C)$$

It is more difficult to justify this assumption for gas phase data and as this work explores different N-containing

functional groups, this nitrate assumption cannot be used. Instead of assuming all N is in the form of nitrate, a range of $OS_N$ (5+, 3+ and 0) are considered and $\overline{OS_c}$ is calculated as a range from a theoretical minimum, where all N is considered nitrate (5+), to a theoretical maximum, where all N is considered cyanate (0). Nitro and nitroso compounds are also considered (3+), but amines and amides are not as they are not expected to be present in the system. As neither the maximum or minimum $OS_N$ is a likely or an accurate description of the system, they merely





represent theoretical limits to calculate $\overline{OS_c}$; the true value will lie somewhere in the middle, in all likelihood between the mean and lower limit as most species are expected not to be cyanates. The $\overline{OS_c}$ reported here is the mean average of three scenarios where all N is considered to exist in 0, 3+ or 5+ oxidation states. This methodology reduces the accuracy of the reported $\overline{OS_c}$ but it clearly and accurately presents the $\overline{OS_c}$ range.

In this document $\overline{OS_c}$ are reported as a function of carbon number. The variation in $\overline{OS_c}$ between all CHON of

the same carbon number is much greater than the variability in $\overline{OS_c}$ of a single CHON compound as a consequence of varying $OS_N$ for that compound.

Where a species has the form $RNO_{1,2}$ the species is considered to be either a cyanate or hydroxy cyanate and the $OS_N$ is set to 0. For $RNO_3$ species the nitrogen is considered to be either a dihydroxy cyanate or a hydroxy $RONO$ or $RNO_2$ compound and so $OS_N$ is set to either 0 or 3+. $RNO_{>3}$ have $OS_N$ set to 0, 3+ or 5+. This leads to the

modification of the $\overline{OS_c}$ parameterisation:

$$\overline{OS_c} = (2 \times O{:}C) - H{:}C - (OS_N \times N{:}C)$$

$$OS_N = 0 \; if \; nO = \leq 2$$

$$OS_N = [0,3] \; if \; nO = 3$$

$$OS_N = [0,3,5] \; if \; nO > 3$$

**2.3.    Hierarchical Cluster Analysis (HCA)**

Hierarchical cluster analysis (HCA) is an analytical technique used to simplify datasets containing large numbers of individual observations into groups or clusters defined by their similarity with the aim of increasing interpretability of the dataset. HCA is an agglomerative technique which allows for the successive merging of individuals or clusters to form larger clusters. Due to the complex nature of atmospheric oxidation and the

detection of many oxidation products by mass spectrometry, it is common practice to reduce the dimensionality of mass spectrometry datasets in order to better describe bulk properties of the process being studied (Sekimoto *et al.*, 2018; Isokääntä *et al.*, 2019; Koss *et al.*, 2019) with HCA being one such method. HCA is independent of calibration as it relies on relative differences between time series' rather than exact concentrations, making it a suitable technique to apply to mass spectrometry data where authentic standards do not exist and individual

calibrations are next to impossible. The final number of clusters is selected based on the distance between them and is determined by the user. Here we use HCA to group the time series' of benzene oxidation products to investigate its utility as a tool for investigating oxidation processes or product properties where the reaction occurs in a continually stirred flow reactor.

In order to use HCA the linkage criterion and distance metric must be selected. The root of the sum of the squares

of a pair of observations (time series), A and B, (i.e. the Euclidean distance) is chosen as the distance metric to define the similarity of the time series' at each time step, t.

$$d_{A,B} = \sqrt{\sum_t (A_t - B_t)^2}$$

The Ward linkage criterion was chosen to determine the distance between sets of observations (U, V, which can be single or multimember clusters) as it gave similar or more interpretable results compared with other linkage



criteria. Here, the sum of the squares of the cluster members ($x_i$ where i iterates over the members U, V or their agglomeration) from their cluster means ($m_U$, $m_V$ or $m_{U \cup V}$) are calculated. This is performed for the two candidate clusters for agglomerating (U, V) and for the new cluster they would form through agglomeration (U∪V). The difference in the sum of the squares between this potential merged cluster and the initial two candidate clusters is the distance used to assess whether the agglomeration of the candidate clusters is acceptable:

$$d_{U,V} = \sum_{i \in U \cup V} \|x_i - m_{U \cup V}\|^2 - \left( \sum_{i \in U} \|x_i - m_U\|^2 - \sum_{i \in V} \|x_i - m_V\|^2 \right)$$

When the increase in the sum of the squares (i.e. distances) is minimal after agglomerating, the two candidate clusters, the new cluster is formed. Here, the time series for all CHO and CHON compounds from the iodide ToF-CIMS measurements were selected for clustering. Time series were scaled between 0-1 to remove the effect of their differing magnitudes and focus on the changes in trends. The scaled time series were then smoothed to a 10

minute average to remove noise that may affect the results of the HCA. This treatment is similar to other studies using HCA with mass spectrometric data (e.g. Koss *et al.*, 2019). The analysis was implemented using the cluster.hierarchy module from SciPy (1.2.0) scientific python library using Python 3.6.

## 3.    Results and Discussion

We detect a range of benzene oxidation products, including many with the same formula as those listed in the

MCM, as well as highly oxidised species with both ToF-CIMS instruments. Well resolved peaks at a given unit mass were frequently observed in the I⁻ spectrum. The separation between the I⁻ ion cluster with its large negative mass defect and the positive defect (excess) of the high H containing deprotonated species allows good peak separation at the resolution of the ToF-CIMS (Fig 2). Beyond I⁻ and $H_2O$, the largest adduct signals we identified with the I⁻ CIMS were low mass species: formic acid ($CH_2O_2$), nitric acid ($HNO_3$), $CH_2O_3$, $C_3H_6O_3$, HONO,

$C_4H_4O_3$ and $C_3H_8O_3$. The prevalence of these ions is likely do to instrument sensitivity rather than chemistry.

### 3.1.    Product distributions

#### 3.1.1.    CHON product distributions: comparison between nitrate and iodide ionisation

Forty-two formulae are found to be common in the low $NO_x$ experiment between the two instruments representing the overlap of oxidation product signals of which 20 are $C_6$ compounds and the rest have lower carbon numbers.

The time series behaviour of the signals at the start of oxidation is highly variable and is specific to each instrument and ion measured by that instrument as oxidant levels and wall partitioning equilibrates. Typically, the level of highly oxidised species increases significantly from the outset but relaxes to lower levels rapidly before they grow again to reach steady state. This early spiking behaviour has been observed in other studies investigating VOC oxidation using the nitrate ToF-CIMS (e.g. Ehn et al. 2014) and is thought to be a consequence of a latent aerosol

sink shifting the equilibrium species from the gas to the aerosol phase. The early spiking also describes the dynamic nature of oxidation concentrations (OH and $O_3$) before the steady state is reached. The iodide CIMS measures a delayed response from wall equilibration processes (Krechmer *et al.*, 2016; Pagonis *et al.*, 2017) making the detection of these rapid changes more challenging. For the range of species measured here, the variability between the two measurements for the same species vary (Fig S1). In general the nitrate observes





square waveforms whereas the iodide observes saw tooth waveforms. Where the agreement is particularly bad it is most likely that different isomers contribute to the different signals measured on the two instruments.

The nitrate ionisation scheme detects a greater number of higher mass species with a higher oxygen content (Fig 3c, 3e) and higher $\overline{OS_c}$ (Fig 3h, 0.84 vs 0.42), whereas the iodide ionisation scheme detects lower mass compounds with a lower oxygen number and lower $\overline{OS_c}$, as has been reported elsewhere in different VOC oxidation systems

(e.g. Isaacman-VanWertz *et al.*, 2017). The average mass of a nitrate adduct (not including the reagent ion) is ~240 g mol$^{-1}$ with ~30% of that attributable to oxygen, whereas the average mass of an iodide adduct is 140 g mol$^{-1}$ with ~20% attributable to oxygen (Fig 3g).

    The nitrate CIMS is able to detect a greater number of high mass species greater than $C_6$ including a large number of dimers ($C_{12}$) whereas the iodide scheme did not detect many species greater than $C_6$ in this instance but did

measure a greater number of low C molecules compared to the nitrate (Fig 3a). Few species greater than $C_{12}$ are detected with either reagent ion, suggesting that in this system, their formation was not likely, as both the nitrate and iodide ionisation schemes are capable of detecting >$C_{20}$ compounds (e.g. Mohr *et al.*, 2019). The iodide scheme detects a maximum number of molecules containing $O_3$ and $O_4$ and very few with more than $O_7$ (Fig 3c). The nitrate scheme observes a broad range of oxygen numbers peaking between $O_9$ and $O_{11}$ but up to $O_{18}$. Iodide

detects an average H:C/O:C of ~1.5 whereas nitrate detects H:C/O:C of approximately 1.1 due to higher O:C ratios (Fig 3f).

With both nitrate and iodide ionisation, the maximum variation in oxygen number per molecule is detected for $C_6$ compounds (Fig 4a) although for nitrate this is closely followed by $C_{12}$ i.e. $C_6$ dimers as well as $C_5$, $C_9$ and $C_{10}$ compounds. The nitrate CIMS detects approximately half the number of low oxygen content ($O_{2-4}$) (and N

containing) oxidation products compared to the iodide CIMS. Subsequently, $\overline{OS_C}$ as a function of carbon number differs between the ionisation schemes at higher carbon numbers (Fig 4b). Below $C_8$ the $\overline{OS_C}$ broadly agree however above $C_8$ the iodide observes negative $\overline{OS_C}$ whereas nitrate observes positive values. This is likely due to the detection of different isomers, which become more prevalent at these at these higher masses.

The saturation concentrations (C*) of the detected species were calculated using the relation of Mohr *et al.* (2019).

The C* indicates that there is significant overlap of SVOC and IVOC detection between the two instruments (Fig 4c). Whilst the iodide ionisation scheme detected HOM, it did not detect any ELVOC (extremely low volatile compounds) and very little LVOC (low volatile compounds), detecting instead mainly IVOC (intermediate volatile compounds), SVOC (semi volatile compounds) and some VOC (Fig 4c). This is in contrast to the nitrate scheme which detected mainly ELVOC, SVOC and some IVOC, although less than the iodide. This highlights

that sampling with both nitrate and iodide schemes captures a large range of gas phase oxidation products, as has been shown in other systems (e.g. Riva *et al.*, 2019).

### 3.1.2. Iodide CIMS detected CHO and CHON under different NO$_x$ conditions

A total of 132 and 195 adduct and deprotonated peaks identified as CHO or CHON in the low and high NO$_x$ experiments respectively using the iodide CIMS of which 126 are common between the experiments (Fig 5). The

number of detected compounds is much higher in the high NO$_x$ case compared with the low NO$_x$ case. This is expected as the complexity of the system increases and additional reaction pathways are viable. This is especially true regarding the detection of N containing compounds. As the chamber had a NO$_x$ background of ~ 300 ppt,



formation of N-containing compounds still occurred during the low $NO_x$ experiment. Twenty four N-containing products were detected during the low $NO_x$ experiment compared with 70 detected during the high $NO_x$

experiment (20 ppb $NO_x$) of which 23 were common.

The majority of species detected are $C_6$ and $C_4$ compounds (Fig 6a) in both cases with $O_{3-4}$ also most abundant. The atom distributions (Fig 6 a-d) appear very similar, with the largest difference being the number of detected N containing compounds (Fig 6 d), where more than twice as many are detected in the high $NO_x$ case. The average carbon and hydrogen atom number for the detected species is marginally higher for the low $NO_x$ case than for the

high $NO_x$ case (Fig 6e). This observation is further enhanced when weighting the average composition by signal rather than merely counting detected species. This is consistent with greater fragmentation occurring under higher $NO_x$ conditions and is also reflected in the average masses of the reaction products when signal weighted (Fig 6g). The O:C and H:C ratios are very similar between the two experiments (H:$C_{lowNOx}$ = 1.34 and O:$C_{lowNOx}$ = 0.93 compared with H:$C_{highNOx}$ = 1.34 and O: $C_{highNOx}$ = 1.02, Fig 6h) and is reflected in $\overline{OS_C}$, which is marginally

higher in the high $NO_x$ case (0.36 vs. 0.42) although the range of possible $\overline{OS_C}$ for both experiments have significant overlap (0.31 - 0.42 for the low $NO_x$ case and 0.28– 0.60 for the high $NO_x$ case, Fig 6h). The small differences in oxidation product characteristics between the two $NO_x$ conditions indicates bulk analysis of all oxidation products is not sensitive enough. In order to better understand the differences between oxidation products from the different $NO_x$ cases, the ions are grouped by preference to formation under high or low $NO_x$

conditions (defined as the regime in which signal enhancement is greatest) and re-analysed (Fig S2). Despite having a very similar average mass, those products that preferentially formed under high $NO_x$ conditions have a lower $\overline{OS_C}$ < 0.13 compared to the low $NO_x$ conditions where $\overline{OS_C}$ ~ 0.39.

In order to better describe these oxidation products, five different groupings of compounds are used to categorise detected oxidation products. Background species refer to those present in the chamber before oxidation and

comprise < 0.05 % of total signal. Compounds of formulae that match species described in the benzene oxidation mechanism of the MCM are denoted MCM and comprise 7.3 % and 6.4 % of the low and high $NO_x$ experiments respectively. Highly oxidised products are identified as species containing 6 or more oxygen atoms and comprise 0.26 % and 0.05 % of the low and high $NO_x$ experiments respectively. These highly oxidised products can include ring breaking and ring retaining products. The rest of the products are termed remainder and are either ring

retaining or ring breaking based on the number of carbon atoms present. Where 6 carbon atoms are present and the double bond equivalent (DBE) equals 4 then the species is ring retaining and where it is less it is defined as a ring breaking product (Mehra *et al.*, 2020). Remaining ring breaking products make up the most of the remaining signal at 89 % and 91 % of the low and high $NO_x$ experiments respectively leaving ring retaining products contributing 3.6 % and 2.5 % to the low and high $NO_x$ experiments respectively.

MCM and ring breaking products are observed to grow continually over the time period of oxidation, whereas highly oxidised and ring retaining products equilibrate and remain flat for the duration of the experiment (Fig 7). This is also true of the high $NO_x$ experiment, although when $NO_2$ photolysis stops halfway through the oxidation, the growth of signal is diminished and the saw tooth profile less apparent. The MCM and ring breaking groups contain small, oxidised molecules that are near the end of the oxidative chain representing large sink of the carbon

in the system and so continually grow. The ring retaining and highly oxidised species reach an equilibrium quickly as they tend to be earlier generation products and so their profiles remain flat.



## 3.2. MCM products

The MCM lists 85 multigenerational oxidation products of benzene including important intermediate and radical species (Jenkin *et al.*, 2003; Bloss *et al.*, 2005). The iodide ToF-CIMS detects 19 and 26 of these oxidation products under low and high $NO_x$ conditions. These signals are a mixture of adducts and non-adduct peaks that match the exact masses of the MCM oxidation products to within 20 ppm error. Whilst only 19 and 26 of the species with a formula cited in the MCM are detected, signals increase for 132 and 195 peaks in the mass spectrum under low and high $NO_x$ conditions respectively indicate many more products are formed than are explicitly accounted for.

The iodide CIMS is able to observe 25 % and 30 % of the listed MCM compounds in the low and high $NO_x$ conditions respectively (Fig 8). The distributions of the number of detected C, H, O and N broadly follows the available MCM C, H, O and N, for example the number of $C_3$ and $C_5$ compounds in this reaction scheme are much fewer than $C_{2,4,6}$ which is reflected proportionally in the number of detected species. This suggests that distributed over the entire mechanism, there is no preference for detection of any specific CHON configurations. A list of all detected species can be found in table S2.

## 3.3. Mechanistic investigation

To test the extent to which the iodide CIMS is able to detect highly oxidised molecules that may originate from autoxidative processes, a number of theoretically suggested formulae based on the autoxidation mechanism with propagation and termination steps were devised and then searched for within the spectra. The applied mechanism includes the benzene oxidation scheme from the MCM with two additional autoxidation steps added in. It consists of autoxidative intra molecular hydrogen shifts from the carbon backbone to a peroxy radical group forming a hydroperoxide group, followed by $O_2$ addition to form a new peroxy radical (Fig 1). The peroxy radical groups were terminated to hydroxyl, peroxyl, nitrate or peroxy nitrate groups, or reduced to form the alkoxy equivalent which then terminates to a carbonyl group or a nitro group (isomeric with nitrite). This was repeated for phenol and catechol precursors providing a total of 53 individual oxidation products with 21 unique formula. We observe 14 and 21 individual deprotonated and adduct signals in the low and high $NO_x$ spectra that correspond to 9 and 11 unique formula respectively. These 9 and 11 unique formula correspond to a maximum of 27 and 33 individual oxidation products (Table S1).

At least one signal (adduct or deprotonated) for all theoretic CHO products are observed in the low $NO_x$ case apart from $C_6H_8O_9$ and $C_6H_8O_{10}$. No derived CHON are observed in the low $NO_x$ case indicating the incorporation of nitrogen into compounds observed under these conditions occurs at a later point. $C_6H_8O_6$, $C_6H_8O_7$, $C_6H_8O_8$ can only be formed through the autoxidation mechanism from either phenol or catechol precursors. All three are found in the low $NO_x$ case and $C_6H_8O_7$ and $C_6H_8O_8$ are found in the high $NO_x$ case. $C_6H_8O_8$ is an exclusively second generation autoxidation product that has been observed previously in benzene oxidation studies (Molteni *et al.*, 2018). $C_6H_8O_9$ and $C_6H_8O_{10}$ are not observed in the high $NO_x$ case. $C_6H_7NO_4$, $C_6H_7NO_6$, $C_6H_7NO_7$, are the only $NO_x$ containing species observed in the high $NO_x$ case. The formation of $C_6H_7NO_4$ has two mechanistic routes, the nitration of the initial hydroxy benzene peroxy radical or the addition of a peroxy nitrate group to the phenol, whereas $C_6H_7NO_6$, $C_6H_7NO_7$ have numerous routes to formation with the only non-autoxidation route being the nitration of phenol and catechol respectively. Exclusively second generation high oxygen content CHON species $C_6H_7NO_8$, $C_6H_7NO_9$, $C_6H_7NO_{10}$, $C_6H_7NO_{11}$ are not observed. Figure 9 summarises these ions in the mass spectra.


### 3.4. Ring retention and ring breaking products

In terms of signal, most of the ring retaining products are $O_{1,2,3}$ of which $O_{1,2}$ is dominated by the formation of phenol and catechol. Conversely, $O_3$ isn't dominated by any one product, but instead comprises of both CHO and

CHON products in both $NO_x$ conditions (Fig 10). Although $O_{1-7}$ species are observed in both cases, higher O numbers of 10, 11 and 12 are only visible in the high $NO_x$ case due to the presence of nitrogen groups that can incorporate more oxygen.

For the ring breaking products, the majority of signal can be attributed to the presence of species containing 3 and 4 oxygen atoms, although at lower carbon numbers of 1 and 2 lower oxygen numbers of 1 and 2 become important

(Fig 11). The majority of signal for both cases is due to $C_{1,2,3}$ compounds. $C_3$ compounds show the greatest signal contribution for the largest range of oxygen numbers. This is especially true in the high $NO_x$ case where signal $C_3$ compounds have a greater contribution from $O_{>4}$. Similarly at higher carbon numbers e.g. 5 and 6, higher oxygen numbers of between 6 and 12 are more readily observed, again where N is incorporated in the high $NO_x$ case a greater inclusion of oxygen is also observed. The form of this organic nitrogen is perturbed throughout the

experiment by altering the $NO/NO_x$ fraction by photolysing $NO_2$.

### 3.5. Effect of $NO_2$ photolysis during oxidation under high $NO_x$ conditions

A four cluster solution was chosen from analysis of the dendrogram (Fig S3) and is sufficient to provide enough detail that can be explained, but not so much that interpretation becomes unclear. Further refinement of clusters greater than 4 offers no more insight in time series behaviour due to oxidation behaviour but rather describes

signal delay likely due to gas wall partitioning (Krechmer *et al.*, 2016; Pagonis *et al.*, 2017). This behaviour is still observed in the four cluster solution, however other affects pertaining to chemistry rather than partitioning are more visible at this low cluster number. Therefore the four cluster solution is chosen (Fig 12). See Table S2. for a list of MCM defined products and their assigned clusters.

Cluster 3 and cluster 4 do not display much effect of $NO_2$ photolysis. Cluster 3 is a slow formation cluster that

grows as photo-oxidation begins. This is likely to represent semi-volatile material that partitions to walls and equilibrates with the instrument lines and IMR. Cluster 4 is a background cluster that decays when photo-oxidation begins (TUV on) and recovers when photo-oxidation stops. These species make up 2.75 % of the total signal during oxidation and so are thought not to contribute majorly to the reaction. The background cluster 4 has the lowest number of $RNO_x$ containing members, and the slow growth, semi-volatile cluster 3 has the greatest number

(Fig 13c). Cluster 4 contains the greatest number of species with high carbon numbers with a negative $\overline{OS_C}$. These characteristics are reflected in cluster 3 but to a lesser extent as carbon numbers are smaller and $\overline{OS_C}$ less negative. It is also notable that cluster 1 products remain at elevated levels in the chamber after all photochemistry has stopped. This again may be indicative of more semi volatile material formed. This is anecdotally corroborated by the higher oxygen and carbon numbers of cluster 1 compared to cluster 2 (Fig 13a). Cluster 2 shares time series

features that are similar to clusters 1 and 3. It has the same long range response as cluster 3 but the same short range increase from photo-oxidation as cluster 1. This is reflected in Fig. 13b where cluster 2 overlaps with clusters 1 and 3, which themselves are well separated. Both these clusters have lower average carbon numbers and higher, positive $\overline{OS_C}$ compared with clusters 3 and 4 (Fig 13b).





Clusters 1 and 2 represent the formation of oxidation products. They increase similarly when photo-oxidation is initiated, however when the $NO_2/NO_x$ ratio is increased, cluster 1 continues to increase but cluster 2 decreases. This suggests cluster 1 products are independent of $NO_2/NO_x$ ratio at these high $NO_2$ fractions; the growth curve is independent of the decrease in NO concentration, suggesting this cluster is characterised by more reactions of $NO_2$ and peroxy radicals. Conversely to cluster 1, cluster 2 decreases when the $NO_2/NO_x$ increases suggesting NO is an important route to formation, either from NO addition products or the increased alkoxy radical fraction

$RO/RO_2$, which is supported by cluster 2 products having higher $\overline{OS_C}$ and lower carbon numbers than cluster 1 products.

Both clusters 1 and 2 contain a similar number of RNO and $RNO_2$ species, but cluster 1 has a greater, odd number of oxygen as $RNO_3$ and $RNO_5$, whereas cluster 2 has a greater even number of oxygen in $RNO_4$ and $RNO_6$. The inclusion of these groupings in one cluster leads to a lack of them in the other; no $RNO_4$ or $RNO_6$ are found in

cluster 1 nor is there a large amount of $RNO_5$ or any $RNO_3$ in cluster 2.

Ideally, NO and $NO_2$ addition to peroxy precursors (cluster 1) would produce nitrates and peroxy acyl nitrates (PANs) that have an odd number of oxygens ($RNO_3$ and $RNO_5$), and addition to alkoxy precursors (cluster 2) would produce nitro- and nitrite compounds that have an even number of oxygens ($RNO_2$). This is indeed observed, however in reality this distinction is not so clear cut as OH addition vs abstraction would reverse this.

So although these observations are broadly followed, they are not exact resulting in a blurring of $RNO_x$ groupings, especially at higher O numbers (Fig 13c). Both clusters 1 and 2 have the same number of $RNO_7$ species despite their time series profiles showing a different dependency on the $NO_2/NO_x$ ratio. It is likely that these $RNO_7$ are structurally different and the oxygen is incorporated into their structures in different ways. For example cluster 1 $RNO_7$, which are all $C_5$ and $C_6$ compounds, may contain more PAN like compounds as these are formed from

peroxy acyl radicals and $NO_2$ addition e.g. to BZEMUCCO3 (MCM), which is the peroxy radical formed from BZEPOXMUC, a first generation ring opening product of benzene, to form BZEMUCPAN ($C_6H_5NO_7$). In contrast, cluster 2 $RNO_7$ are smaller $C_3$ and $C_4$ compounds, which may require the fragmentation of larger organic precursors enhanced by the presence of NO e.g. the unimolecular decomposition of the alkoxy radical MALANHYO to form HCOCOHCO3 and then HCOCOPAN ($C_3H_3NO_7$).

**4. Conclusion**

Two ToF-CIMS using iodide and nitrate ionisation schemes were deployed at the Jülich plant chamber as part of a series of experiments examining benzene oxidation by OH under atmospherically relevant high (20 ppb) and low (0.3 ppb) $NO_x$ conditions.

Both ionisation schemes detect benzene oxidation products including highly oxidised organic molecules. Nitrate

CIMS detects many $C_{12}$ dimers and a greater number of species with high oxygen ($O_9$-$O_{11}$). This translates to higher $\overline{OS_C}$, especially at higher carbon numbers ($C_{\geq 8}$), and lower C* indicating the detection of ELVOC, SVOC and IVOC. In contrast, the iodide CIMS detects no dimers, but many more monomers and ring breaking products ($C_{\leq 6}$) than the nitrate scheme, with most common oxygen numbers of ($O_3$-$O_4$). The $\overline{OS_C}$ of high carbon species is lower, although at lower carbon numbers ($C_{<6}$) $\overline{OS_C}$ between the two ionisation schemes broadly agree due to the

increased likelihood of measuring the same species, rather than isomers. The corresponding C* measured by the



iodide CIMS suggests no ELVOC was detected, but many more SVOC, IVOC and some VOC were measured when compared to the nitrate CIMS. Thus the two ionisation schemes cover a large range of volatilities.

132 and 195 CHO and CHON species are detected in the low and high $NO_x$ experiment respectively of which 126 are common. In both cases these are mostly $C_4$ and $C_6$ compounds. A greater number of oxidation products are

measured in the high $NO_x$ case, including 70 N containing compounds compared to 24 in the low NOx case, of which all but one are common. Splitting oxidation products into five categories, the contribution to signal increases sequentially from: background ($< 0.05\,\%$), highly oxidised products ($0.05 - 0.26\,\%$), ring retaining ($2.5 - 3.6\,\%$), MCM ($6.4 - 7.3\,\%$), and ring breaking ($89 - 91\,\%$). The $6.4 - 7.3\,\%$ of signal assigned to MCM products represents $25 - 30\,\%$ of the 85 listed multigenerational oxidation products detected under low and high $NO_x$

conditions and are proportionally distributed across all carbon numbers. Highly oxidised and ring retaining products equilibrate quickly within the system and remain throughout the experiments, whereas MCM and ring breaking products continue to grow throughout the experiments. Highly oxidised and ring retaining products are earlier generation products which form quickly and equilibrate as loss and formation processes equalise. MCM and ring breaking products contain smaller, more oxidised molecules that are further down the oxidation chain

and represent the largest destination of carbon within the system. Signal from ring retaining products is dominated by phenol ($O_1$) and catechol ($O_2$) whereas $O_3$ compounds comprise a number of different species. Ring breaking products are dominated by $C_1$, $C_2$ and $C_3$ compounds with similar oxygen numbers. $C_3$ compounds show the greatest variability of oxygen atom number. For both retention and breaking products, high O numbers of up to 7 are identified, however in the high $NO_x$ case where incorporation of $RNO_x$ is greater, higher O numbers of 10, 11

and 12 are observed. Within the context of the theoretical mechanistic investigation (section 3.3), iodide ionisation is able to detect 27 and 33 species belonging to potential autoxidation reaction pathways. These detected species include one previously observed, exclusively second generation autoxidation product $C_6H_8O_8$ and all derived 1st generation autoxidation products. However it is noted that only two of these first generation products steps ($C_6H_8O_6$, $C_6H_8O_7$) are formed exclusively through autoxidation whilst the rest have other routes to formation.

Higher oxygen content products $C_6H_8O_9$ and $C_6H_8O_{10}$ are not observed in either case.

Clustering the time series in the high $NO_x$ experiment into four clusters distinguishes two clusters that contain products formed from photo-oxidation. One of these clusters (cluster 1) is independent of the $NO_2/NO_x$ ratio whereas a second cluster (cluster 2) decreases, as a result of NO dependent formation, either through addition and/or an increased $RO/RO_2$ fraction. Cluster 2 has higher $\overline{OS_C}$ and lower carbon numbers than cluster 1

suggesting it consists of more oxidised and fragmented compounds consistent with an increased $RO/RO_2$. Cluster 2 contains $RNO_4$ and $RNO_6$ but no $RNO_3$ and little $RNO_5$ whereas the opposite is true for cluster 1. This somewhat agrees with theoretical $RNO_x$ product distributions as $NO_x$ addition to alkoxy radicals (cluster 2) is more likely to produce even oxygen content $RNO_x$ (through the formation of nitrites and nitro compounds), and odd oxygen $RNO_x$ through addition to peroxy radicals (cluster 1, such as nitrates and PANs). For species with larger oxygen

content e.g. $RNO_7$ which is detected in both clusters, the carbon number is lower for cluster 2 ($C_{3,4}$) compared to cluster 1 ($C_{5,6}$) indicating more fragmentation has occurred, again implying a greater contribution from the alkoxy channel. It is noted that the effect of OH addition rather than H abstraction as the initiation step to the reaction will reverse this pattern, and along with other unimolecular rearrangements, may explain some of the $RNO_x$ cluster variability observed.



## Contributions

MP analysed nitrate and iodide ToF-CIMS data and wrote the manuscript. OG analysed nitrate ToF-CIMS data and provided peaklist data. TJB and MLB operated the iodide CIMS. DES, AM, SDW and AB contributed to iodide CIMS analysis with oversight of CJP and HC. CJP and GM led the development of the experiments presented here. TFM, AKS, CJP, MH, GM, ÅH designed the experiments. SK, IP, SS, RT, EK, DZ, JW operated the nitrate CIMS and performed the experiments.

## Acknowledgements

This work was conducted during a PhD study supported by the Natural Environment Research Council (NERC) EAO Doctoral Training Partnership and is fully-funded by NERC whose support is gratefully acknowledged (Grant ref no. NE/L002469/1).The study was funded by European Research Council grant 638703 and supported by Doctoral School in Atmospheric Sciences at the University of Helsinki (ATM-DP). This work was supported by the Swedish Research Council (grant number 2014-5332). Å.M.H. acknowledges Formas (grant number 214-2013-1430) and Vinnova Sweden's Innovation Agency (grant number 2013-03058), including support for her research stay at Forschungszentrum Jülich. Part of the research in this study was performed at the Jet Propulsion Laboratory, California Institute of Technology, under a contract with the National Aeronautics and Space Administration.

## Competing Interests

The authors declare no competing interests.

## Data Availability

Data is available on request.

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

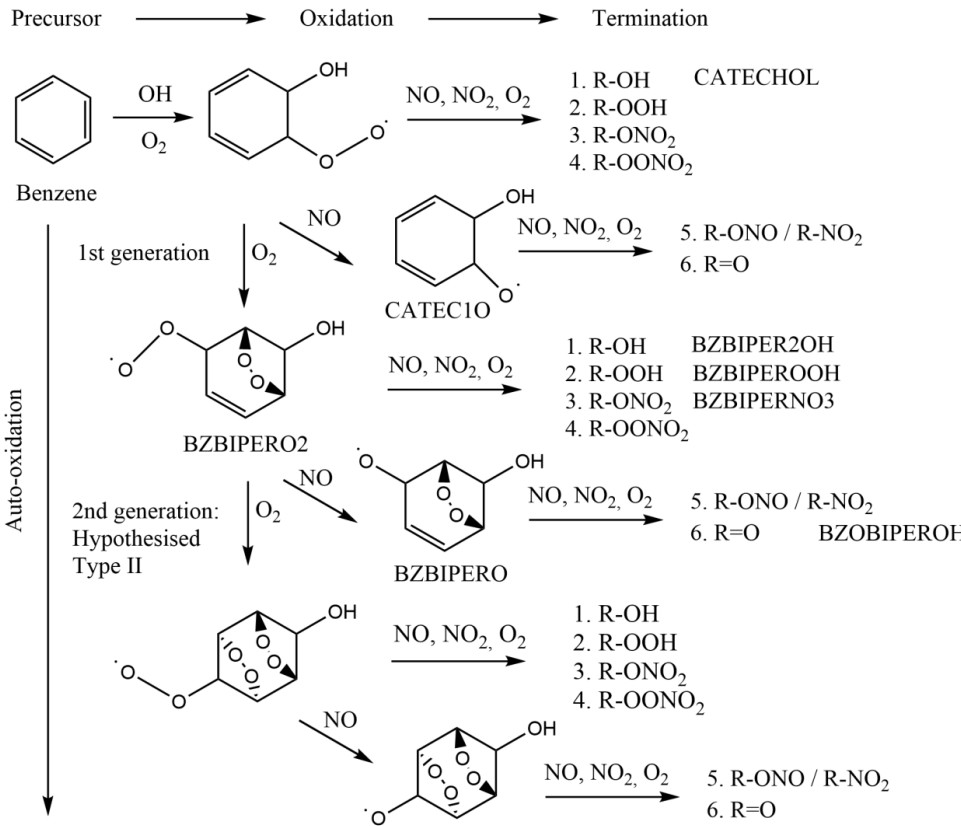

Figure 1. Reaction scheme of benzene oxidation by OH with proposed di-bridged species adapted from Molteni *et al.*, (2018). Species present in the MCM are labelled.





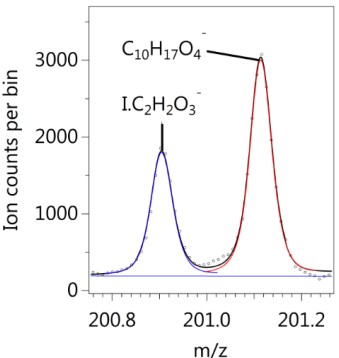


Figure 2. Example of peak separation between an iodide adduct I.C$_2$H$_2$O$_3^-$ and deprotonated signal C$_{10}$H$_{17}$O$_4^-$ detected during background measurements before the low NO$_x$ experiment.

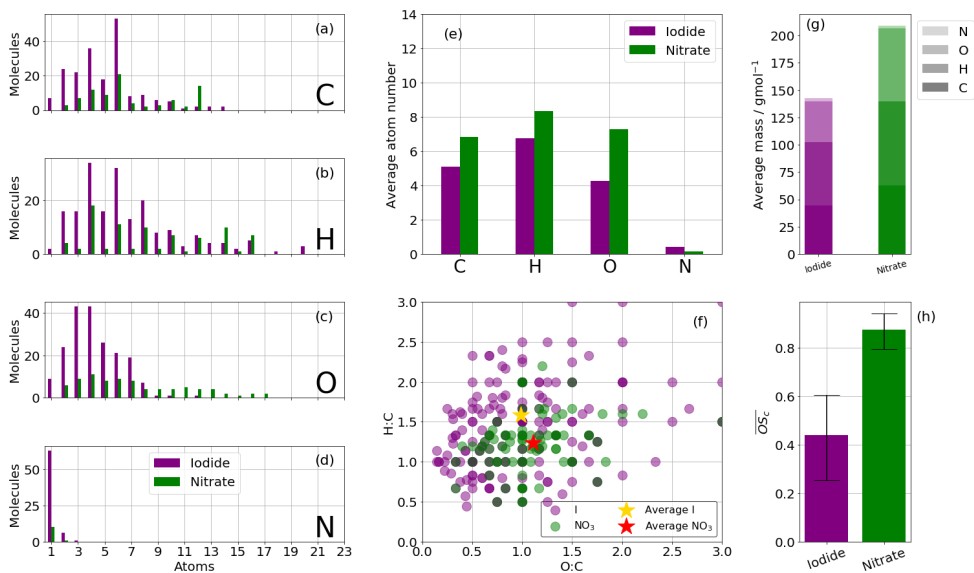

Figure 3. Summary of CHO and CHON statistics of detected oxidation products during the high (20 ppb) NO$_x$ benzene

oxidation for iodide (purple) and nitrate (green) ionisation schemes. (a, b, c, d.) Frequency distributions of the atoms C, H, O and N for the detected oxidation products. (e) Average number of atoms per detected oxidation product split into C, H, O and N. (f) Van Krevelen diagram (O:C vs H:C). (g) The average mass of a detected oxidation product detected by iodide and nitrate ionisation. (h) The average carbon oxidation state $\overline{OS_C}$ of oxidation products detected by iodide and nitrate ionisation. Limits represent minimum and maximum $\overline{OS_C}$ as a function of minimum and maximum OS$_N$, see section 2.3.





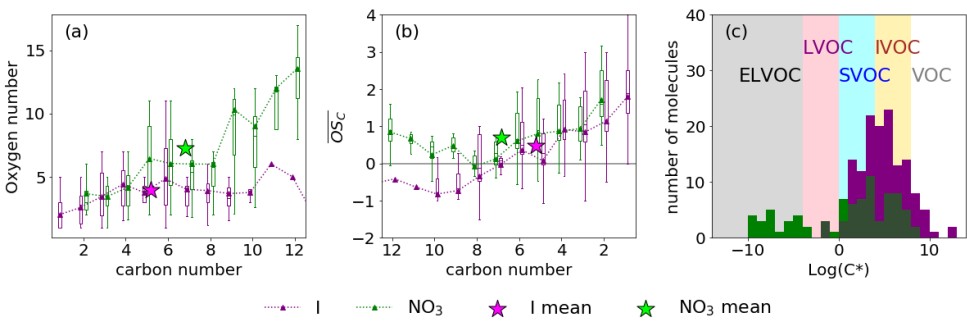


Figure 4. Comparison of organic oxidation products from the iodide (purple) and nitrate (green) ionisation schemes. (a) carbon number vs oxygen number. (b) average carbon oxidation state ($\overline{OS_C}$) vs carbon number. (c) Saturation concentration (C*) based on Mohr *et al.* (2019).

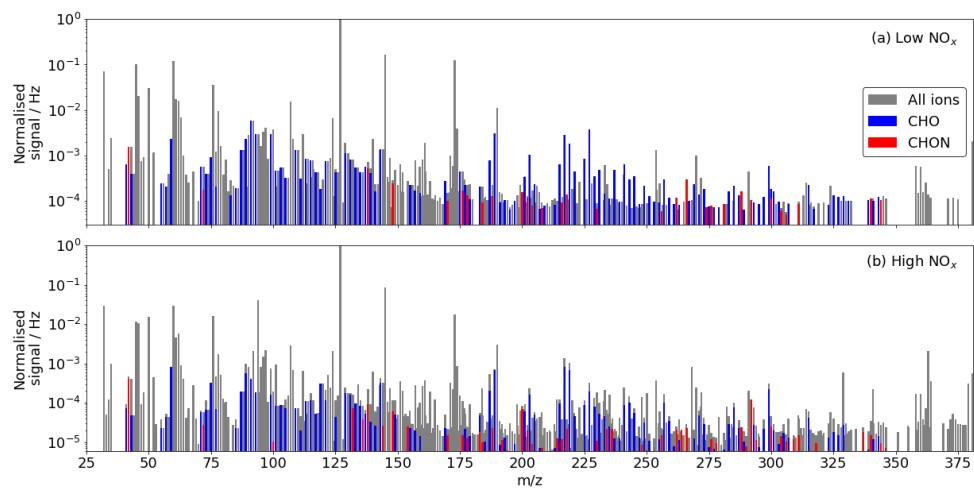


Figure 5. (a) Unit mass spectrum during benzene oxidation by OH under (a) low $NO_x$ conditions and (b) under high $NO_x$ conditions. Identified CHO and CHON compounds are highlighted.

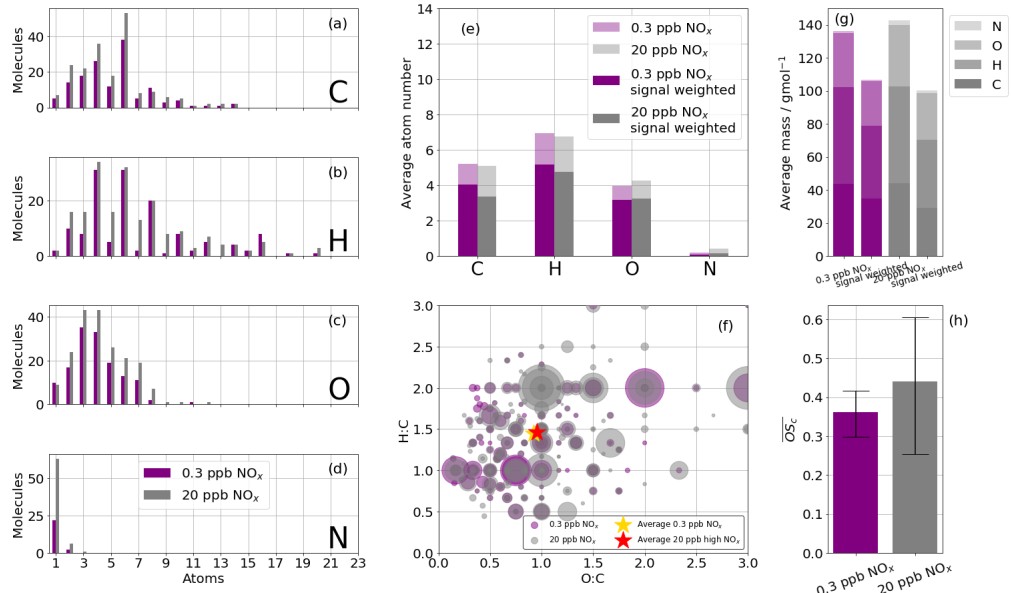

Figure 6. Summary of CHO and CHON statistics of detected oxidation products during high (20 ppb, grey) and low (0.3 ppb, purple) NOx benzene oxidation with the iodide ionisation scheme. (a, b, c, d.) Frequency distributions of the atoms C, H, O and N for the detected oxidation products. (e.) The average number of atoms per detected oxidation product split into C, H, O and N atoms, also shown is signal weighted average atom number. (f.) Van Krevelen diagram (O:C vs H:C) sized by signal. (g.) The average mass of a detected oxidation product for high and low NOx conditions, also shown signal weighted average masses. (h.) The mean, average carbon oxidation state ($\overline{OS_C}$) of the detected oxidation products for high and low NOx conditions. Limits represent minimum and maximum OSC as a function of minimum and maximum OSN, see section 2.3.

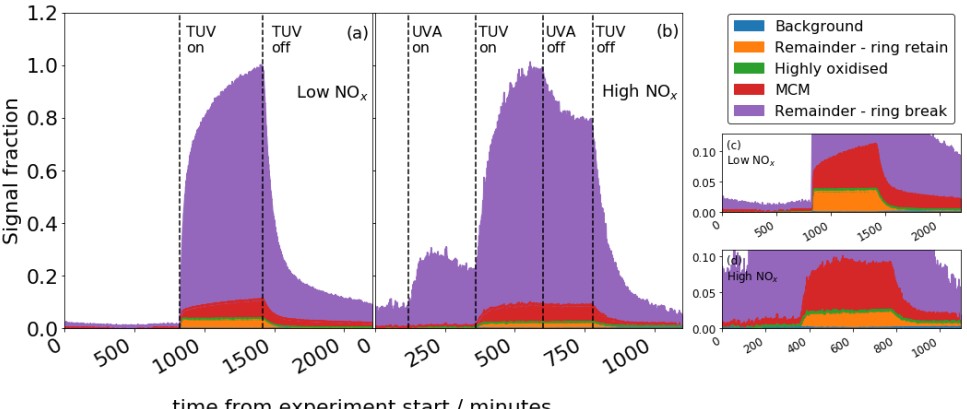

Figure 7. Total signal from CHON + CHO compounds for the low NOx (a) and high NOx (b) experiments. (c) and (d) show the same data in (a) and (b) with a reduced y scale; y and x scale have the same units.



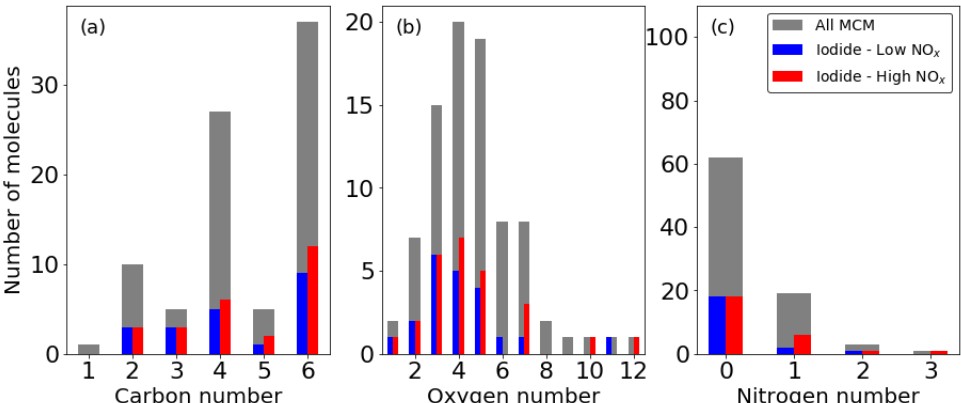

Figure 8. Fraction of ions observed from the MCM in the high and low $NO_x$ experiments expressed as number of molecules containing atom numbers for (a) carbon, (b) oxygen and (c) nitrogen.

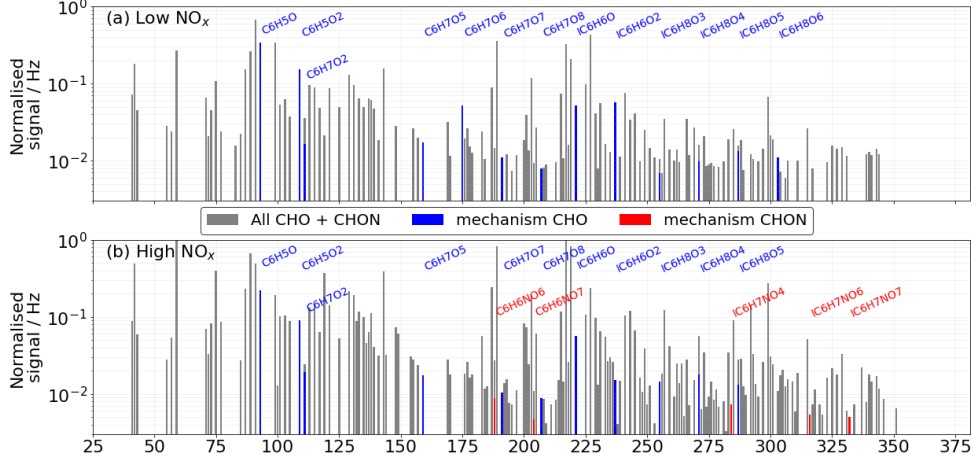

Figure 9. Unit mass spectra of (a) low $NO_x$ and (b) high $NO_x$ experiments. Detected ions from the mechanism are highlighted as CHO (blue) or CHON (red).





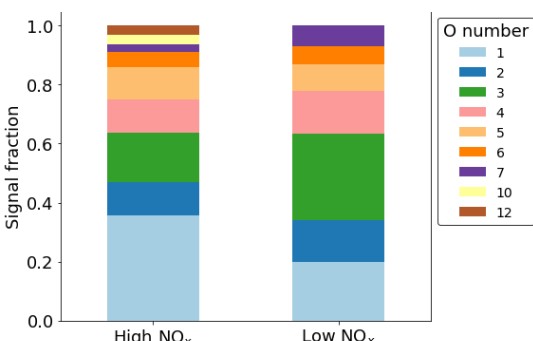

Figure 10. Normalised signal fraction of ring retaining products under high and low $NO_x$ conditions. Signal for phenol and catechol dominate for both conditions and make up the $O_1$ and $O_2$ fractions.

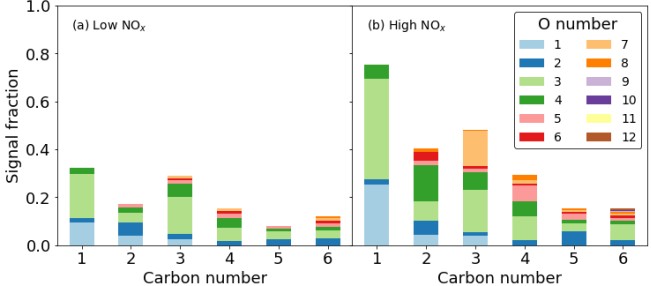

740  Figure 11. Signal fraction as a function of carbon number for ring breaking products for the (a) high $NO_x$ and (b) low $NO_x$ conditions.

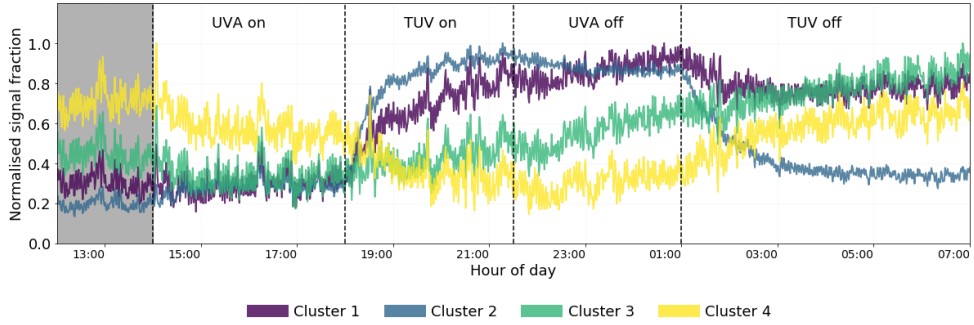

Figure 12. Time series of the four cluster means for the high $NO_x$ experiment. Clusters 3 and 4 represent slow formation and background clusters and give no information regarding the effect of $NO_2$ photolysis. Clusters 1 and 2 display similar
745  behaviours when the TUV light is switched on, but different behaviours when the UVA light is switched off, indicating their members are sensitive to the change $NO/NO_x$ ratio determined by the UVA light state. The shaded area is the dark state before any photolysis occurs.

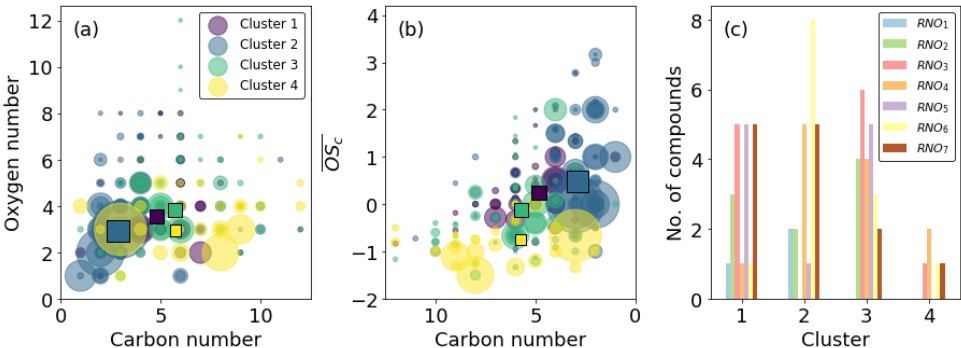

Figure 13. Cluster characteristics (a) carbon vs oxygen number (b) carbon number vs average carbon oxidation state. Squares
indicate cluster averages. (c) Number of RNO$_x$ species. Clustering based on time series similarity provides good separation
in $\overline{OS_C}$ vs C space (b) and splits RNO$_{odd}$ and RNO$_{even}$ in clusters 1 and 2 (c).

750