# Peer review of "Chemical characterisation of benzene oxidation products under high and low $NO_x$ conditions using chemical ionisation mass spectrometry"

_Atmospheric Chemistry and Physics, 2020_

## Referee Comment (RC1) · Anonymous Referee #1 · 21 Sep 2020

Review of Priestly et al. "Chemical characterisation of benzene oxidation products under high and low NOx conditions using chemical ionisation mass spectroscopy."

Synopsis of the manuscript. This study examines the products from the photooxidation of benzene in the presence and absence of nitrogen oxides (NOx) presumably with application to the ambient atmosphere. Mixtures of benzene, ozone, and water vapor in air with or without NOx are irradiated in a 1.45 m3 borosilicate glass chamber operated as a continuous stirred tank reactor (CSTR). For benzene to react in this relatively small environmental chamber, a robust source of hydroxyl radicals (OH) is

needed. This is achieved through the photolysis of ozone at 254 nm giving O(1D) with then reacts with water present in the system to generate OH, although the photolysis of oxygen is also possible via the 185 nm Hg line. Once the system comes to steady-state at a chamber residence time ($\tau$) of 2900 s, the gas-phase products are measured via a chemical ionisation mass spectrometer (CIMS) using I- and NO3- as the reagent ions. Multiple analysis methods to measure products, comparison with the Master Chemical Mechanism (MCM) model, a comparison of ring-fragmentation and ring retaining products, oxidation state analysis (OSA), and hierarchical cluster analysis (HCA) are used to interpret the CIMS product data. From the high resolution time-of-flight (TOF)-CIMS, empirical formulae for the ion peaks can be obtained with reasonable confidence which are then compared with stable MCM products and dozens of coincidences are observed. The OS developed almost exclusively for particle measurements (Kroll et al., 2011) are adopted for this work. The findings are all qualitative in that calibration factors are unattainable for the compounds of interest in this work. The interpretation includes findings for the volatility classification of the product distribution, differences in the importance of ring-fragmentation and ring-retaining products, and differences in the role of NOx in the mechanistic system.

Overall impressions. The findings are interesting and potentially valuable given the importance of benzene in the atmosphere. Moreover, the sophisticated instrumentation being used allows data to be obtained, which heretofore has been nearly impossible. The mechanisms of aromatic systems are extremely important for understanding both ozone and particle mass formation. Thus, the paper has great potential in adding to the body of mechanistic knowledge needed to predict these species. The paper is well written and reasonably well organized.

However, while the description of the findings in the abstract seems impressive, confidence in them quickly deteriorates upon further reading. For me that occurs at the beginning of Section 2. First, for the type of product study being undertaken, a 1.45 m3 reaction vessel is extremely small, which, of course, requires that the system be operated as a CSTR – a requirement more than just a means to homogenise the mixture. Since it was not stated, the vessel was probably uncoated and the OH wall loss must have been substantial and undoubtedly diffusion limited. This is an infinite sink thus, representing a constant gradient of OH in the system. Now we come to the production of OH. Creating OH from the photolysis of ozone at 254 nm followed by $O(1D) + H2O$ strikes me as extremely dangerous for interpreting the data. Of course, the radiation is not only available to photolyze ozone but the other organics and the various nitrogen oxides in the chamber. Benzene has a strong absorption at 254 nm and likely being photolysed, depending on the quantum yield. The steady-state concentration of ozone is thus very important. Clearly, ozone does not react with benzene, but once fragmentation occurs, there are plenty of double bonds for ozone to react with, perhaps not as competitively as OH, but that only depends on the concentration of ozone being used. This leads to my first major complaint. Please give the concentrations or concentration ranges being used in the study. Better yet a table of the initial conditions: benzene, ozone, water (ppm), NO, NO2, chamber residence time is essential. A table of initial conditions should not be relegated to a supplementary section. It is ACP after all; there is no page limitation.

Onto the OH concentration in Table 1. I simply find it implausible that OH concentrations of 10(+8) molecules can be formed in what is basically a smog chamber. (Obviously, high concentrations of OH are generated in flowtubes operated on millisecond timescales. Not here.) There are just too many things for OH to do. As mentioned, walls and double-bonds are but two. This is all based on the p.5 equation for benzene loss and the low rate constant for irradiations over a period of minutes results in high calculated OH levels. First, I would call Column 3, benzene loss because that is what is measured. Second, the absolute benzene loss is irrelevant. Thus, either the initial benzene concentration in Table 1 caption or percent loss in the table should be given. In my opinion, the observed loss is some combination of reaction with OH and photolysis by 254 nm mercury radiation. This would thus lower the calculated OH concentration. However, if we take these concentrations seriously recognizing that the products are

likely to be considerably more reactive than the parent compound, it is easy to believe that 3-5 generations of products are likely to be present in the chamber with the CIMS measuring all of them, and perhaps with increased oxidation, later generation products are being measured more sensitively than earlier generation products.

Of course, a major issue as noted in the manuscript is a lack of calibrations for products formed. This is unavoidable in these types of experiments. This is even worse when using a CIMS since the sensitivity of seemingly similar compounds can be one or more order-of-magnitude different. Therefore, to compare signal strength, as though implying, they are linearly related to concentration is a rapid path to misinterpretation. I would recommend that graphics based on signal strength should be treated very carefully and worded conservatively.

The comparison with MCM I find both tricky and the reverse of the experimental-deterministic model paradigm. First, given the complexity of MCM and the scarcity of quantitative data upon which later generation (beyond first) mechanisms are based I would describe the mechanisms at best as being tentative. It would be better to use the data in these experiments to provide some credence for later generation MCM product predictions, of course, not stating anything about predicted yields. Otherwise, it looks like "the blind leading the blind".

Something needs to be added regarding the possible role of NO3 radicals in the system, especially given the number of oxidized nitrogen compounds being detected. Given what I expect to be a high ozone concentration (ppm), its reaction with NO2 is probably occurring to some non-negligible degree. I see no evidence provided that autooxidation plays an important role in the system. Perhaps, it does, but not from the data presented.

Of course, it may be that the authors have considered and addressed the issues above and similar issues. However, these issues are critical to understanding how the data are interpreted and must be included in the paper, if the study is to have any credibility.

Finally, I believe it would be valuable to show how the findings from this paper fit into the understanding of benzene oxidation. Thus, it is my opinion that this manuscript needs substantial work before it is published.

A few line-by-line comments.

Line 77. Johnson et al. 2005 is a rather poor reference. Dicarbonyl aldehydes were shown to be formed in aromatic systems by the mid-1980's (e.g., Dumdei and O'Brien, Science, 1983; Shepson et al. JPC, 1984; Dumdei et al, EST, 1988.)

Line 132. I am not sure what this sentence is saying. Mass closure, if that is ever possible, will require a lot more than complementary instruments. In fact, as these experiments are conducted, each product generation probably gets you further and further away from "mass closure". I would reword the sentence in a more realistic fashion and perhaps with a bit less jargon.

Line 148-49, Table 1. Is it true that the reacted benzene concentrations were identical to three significant figures for the experiments shown? The actual measured data would be more helpful together with the percent benzene loss.

Line 153. The sentence beginning in this line is a "red herring". Otherwise, reactions of benzene with ozone in the chamber would lead to product distributions being a complete mess. If benzene and ozone were introduced together, there would probably be less of a mixing issue.

Line 204. How about calibrations for the instrument sensitivity for the inorganic nitro-geneous compounds? Were any undertaken?

Line 246. I am not sure what the term "time series" means in the context of these experiments. They were all done under steady-state conditions.

Line 275. This would be an excellent place to include a table for the initial conditions. This would also allow us to see how many experiments the data is based on. One of my main questions is how many different residence times were tested. For example, it

would have been helpful to have one experiment at $\tau$=1450 s and another at $\tau$=5800 s. (Note: I am not asking for any additional experiments if only one residence time was tested.)

Line 280. Again, I am not sure what "time-series behavior" means when conducting experiments in a steady-state reactor.

Line 283. The other studies mentioned were probably performed with the experiment(s) being conducted in a batch mode. However, for a CSTR used in reactive systems, a broad spike occurs at the beginning solely from the dynamics of a reacting system reaching steady state. Thus, three or four residence times are typically needed to achieve steady state. Moreover, I suspect the system is not fully homogeneous and certainly not at the beginning of the irradiation. OH is certainly a problem – high concentrations near the lamp and low concentrations near the wall. There is also the question as to whether the radiation is optically thin. I believe there are many factors leading to this "spiking" with gas-aerosol partitioning probably being a minor one.

Line 381. I find this section a bit too speculative and should be written more conservatively. I again repeat my comments regarding MCM and the CIMS data. Certainly, the CIMS data has no quantitative significance, since the sensitivities are typically all-over-the-map. This section would be a good place to consider what the uncertainties are present in these experiments.

Line 421. A consideration of the photolysis of the nitrogenated products at 254 nm might be considered in this section as well.

Line 470. The first sentence of the conclusion is totally unsupported. The sentence reads as though NOx is decoupled from OH. Experiments having steady-state OH concentrations near 10(+8) molec/cc have at best marginal relevance even with atmospherically plausible levels of NOx being tested. I would be far more circumspect in making atmospherically relevant statements from this dataset. At minimum, additional qualifying statements need to be provided in this paragraph. I would highly recommend

simply leaving out the first paragraph.

Line 486ff. From this point on, I think the data is being overinterpreted.

Whilst it may be trivial, "whilst" strikes me as being rather anachronistic, perchance appropriate in poetry, less so in scholarly writing.
* * *

---

## Referee Comment (RC2) · Anonymous Referee #2 · 24 Sep 2020

This manuscript reported the measurements of benzene oxidation products with two CIMS using I- and NO3- as reagent ions. The discussions focus on (1) difference in products measured by I- vs NO3-; (2) difference in produces between one low-NOx experiment and one high-NOx one; (3) detected products vs the ones in MCM. The measurements are performed with the state-of-the-art instruments and the analysis in the manuscript is solid. However, the major issue is that the discussions are fragmented. Many interesting observations are presented, but it is a bit blurred how such detailed measurements of nearly 200 ions improve our understanding on the funda-

mental oxidation mechanism of benzene, besides leaving the impressions that the benzene oxidation generates hundreds of products and the oxidation products are different between low- and high-NOx conditions. The manuscript can be largely improved if the detailed discussions can be synthesized in a coherent fashion. Overall, I recommend publication after major revision.

Comments 1. Many products more than 6 carbons (like C8, C9) have been detected. Please discuss the potential formation mechanisms of these compounds. 2. Some comparisons between measurement and MCM are not conducted in a meaningful way. For example, in Page 10 Line 351, it is claimed that compounds of formulae that match species in MCM compare 7.3% and 6.4% of the low and high NOx experiments, respectively. There are two issues in this comparison. First, the values depend on the extent of oxidation. For example, it is well-studied that the phenol yield in benzene oxidation $\sim$50%. MCM compounds should least comprise 50% of detected compounds when the secondary chemistry is negligible. Second, as the response factors in I-CIMS vary by orders of magnitude, the raw signal in Hz cannot represent the true product distribution. 3. Similarly, because of the two issues mentioned above (i.e., uncertainties in instrument sensitivity and extent of oxidation), it is unclear how meaningful the reported distribution of products is. The discussion on Page 10 Line 359 is one example. 4. Page 11 Line 375. How are these two values calculated? 5. Page 11 Line 396-397. This statement on the potential formation mechanism of $C_6H_8O_6$, etc is too strong. The HOMs formation mechanism from benzene oxidation is unclear. For example, Garmash et al. 1 showed that the HOMs yield is higher in benzene oxidation than phenol oxidation. It lacks support to state that $C_6H_8O_6$, etc can only be formed from phenol or catechol. 6. Table 1. Please include the initial concentration of benzene. I estimate that roughly 20-40% of initial benzene is oxidized in the experiments. Because benzene oxidation products are much more reactive than benzene, many detected products are likely from multi-generation chemistry. This should be clearly mentioned in the manuscript.

Reference 1. Garmash, O.; Rissanen, M. P.; Pullinen, I.; Schmitt, S.; Kausiala, O.; Tillmann, R.; Zhao, D.; Percival, C.; Bannan, T. J.; Priestley, M.; Hallquist, Å. M.; Kleist, E.; Kiendler-Scharr, A.; Hallquist, M.; Berndt, T.; McFiggans, G.; Wildt, J.; Mentel, T. F.; Ehn, M., Multi-generation OH oxidation as a source for highly oxygenated organic molecules from aromatics. Atmos. Chem. Phys. 2020, 20 (1), 515-537.
* * *

---

## Author Comment (AC1) · 2 Nov 2020

- 1 General response to Reviewers 1 and 2.
- We thank the reviewers for taking the time to review this manuscript and note their comments have
- 4 improved the overall quality of the work. The original comments are left in black, the response is given in
- 5 blue and the updates to the manuscript are red.

**Reviewer 1**

7

**8 General point.**

Overall impressions.

12 The findings are interesting and potentially valuable given the importance of benzene in the 13 atmosphere. Moreover, the sophisticated instrumentation being used allows data to be obtained, which heretofore has been nearly impossible. The mechanisms of aromatic systems are extremely 14 15 important for understanding both ozone and particle mass formation. Thus, the paper has great 16 potential in adding to the body of mechanistic knowledge needed to predict these species. The paper 17 is well written and reasonably well organized. However, while the description of the findings in the 18 abstract seems impressive, confidence in them quickly deteriorates upon further reading. For me that 19 occurs at the beginning of Section 2. 20

First, for the type of product study being undertaken, a 1.45 m3 reaction vessel is extremely small,
which, of course, requires that the system be operated as a CSTR – a requirement more than just a
means to homogenise the mixture.

We do not understand the concerns of the referee. Why is the vessel "extremely" small? And what is wrong about using continuously stirred tank reactors for product studies? We assume there must be a misunderstanding because we did not express the performance as CSTR explicitly enough. From the referee remarks here and later below, it seems that we did not state clearly that we actively stirred the reactor.

- 30 We will now more clearly state in the manuscript, that
  - JPAC is operated as continuously stirred tank reactor (and not as a simple flow reactor)
  - a Teflon fan provides typical mixing times of less than 2 minutes, which compares to residence times of 48 min.
  - the reactor was run in steady state which leads to stable conditions, which are determined by the continuous inflow of the reactants, chemical production and destruction of products, wall losses, and of course the outflow.
    - we measure the steady state concentrations by measuring in the outflow.

38 39

32

Since it was not stated, the vessel was probably uncoated and the OH wall loss must have been substantial and undoubtedly diffusion limited. This is an infinite sink thus, representing a constant gradient of OH in the system. Now we come to the production of OH. Creating OH from the photolysis of ozone at 254 nm followed by O(1D) + H2O strikes me as extremely dangerous for interpreting the data. Of course, the radiation is not only available to photolyze ozone but the other organics and the various nitrogen oxides in the chamber. Benzene has a strong absorption at 254 nm and likely being photolysed, depending on the quantum yield.

- We provide the following clarification of the OH budget within JPAC and discussion on benzene
  absorption at 254 nm. The reactor is operated as CSTR and measurements are taken when the
  system is in steady state. The chamber is actively mixed within 2 minutes. This leads to a well-mixed
  core and a boundary layer, which must be penetrated by diffusion before radicals and low volatility
  compounds are eventually lost to the wall.
- 54 We estimate the typical time for moving by Brownian motion through a layer thickness I:
- 55

 $t_{diff} = \frac{l^2}{D}$ 56 57  $l = \sqrt{D \cdot t_{diff}}$ 58 59 60 D is a diffusion coefficient. For a typical HOM molecule  $D \approx 0.05$  cm2/s, HOM lifetime 100 s => I = 2 61 62 cm  $\Rightarrow$  We estimate the "effective" boundary layer thickness is a few mm. 63 64 65 OH background "reactivity" has been determined in collaboration with our LIF group (Broch, thesis, 2011). OH background loss over the years has been pretty stable at 4 s-1, including wall losses. 66 67 ⇒ Background OH Lifetime 0.25 s. If we assume a diffusivity of OH = 0.3 cm2/s (larger than water = 0.25 cm2/s but smaller than H2 = 0.6 68 cm2/s) and laminar layer of 5 mm then 69 70  $0.83 = \frac{0.5^2}{0.3}$  [s] 71 72 73 OH wall loss coefficient = 1.2 s-1 74 That means OH wall loss contributes about  $\frac{1}{4}$  (1.2 s-1) to the OH background loss. 75 76 When we do experiments with VOC we have to also to consider the lifetime of OH due to VOC 77 oxidation. 78 Here, the lifetime of OH with respect to benzene oxidation: 79  $k_{OH+C6H6} = 1.210^{-12} \text{ cm}^{-3} \text{ s}^{-1}$ benzene = 50 ppb = 1.2510-12 cm-3 80 81  $\tau_{OH+C6H6} = \frac{1}{1.2 \cdot 10^{-12} \cdot 1.25 \cdot 10^{12}}$ 82 83 ⇒ TOH+C6H6 = 0.7 sec. This number is likely even shorter due to secondary oxidation of reaction 84 products, so we estimate it to be about half of that, 0.3 s 85 Thus, the estimation of the total OH loss becomes 86 87  $7.3[s^{-1}] = \frac{1}{0.25[s]} + \frac{1}{0.3[s]}$ 88 89 90 This overall OH loss rate can be rounded up to 10[s-1], 91 92 Our claim is that we are able to reach OH  $\approx 10^8$  cm-3 under steady state conditions, in other words the 93 destruction of OH =10x108 cm-3 s-1 94 95 In steady state: 96 production = destruction 97 To keep up 1.108 cm-3 OH in steady state, the production POH has to be 98 99  $P_{OH} = 10 \cdot 1 \cdot 10^8 \text{ cm}^{-3} \text{ s}^{-1}$ 100 which means that to keep up a steady state concentration of 108 cm-3 s-1 OH, a production of 109 OH 101 cm-3 s-1 is needed. 102 103

OH is produced in the chamber by photolyzing  $O_3$  into  $O^1D$ , and the fraction of  $O^1D$  reacting with 105 water to form 2 OH radicals is in competition with quenching of the  $O^1D$  into  $O^3P$  by  $N_2$  and  $O_2$ 106 (=0.07).

$$P_{OH} = \frac{2 \cdot J_{01D} \cdot O_3 \cdot 2.14 \cdot 10^{-10} \cdot H_2 O}{3.2 \cdot 10^{-11} \cdot \text{EXP}(67/\text{T}) \cdot O_2 + 2.0 \cdot 10^{-11} \cdot \text{EXP}\left(\frac{130}{\text{T}}\right) \cdot N_2 + 2 \cdot 10^{-10} \cdot H_2 O}$$

110The water concentration at 288K and  $\approx 60\%$ RH =  $3x10^{17}$ . The J01D in the chamber is varied by111shielding parts of the quartz lamps with glass tubes.112113113In many experiments the J01D was  $2.610^{-3}$  s-1 for the whole chamber, but it can be enlarged up to114 $5.210^{-3}$  s-1. At steady state concentrations of  $1x10^{12}$  cm-3 (40 ppb) of O3, we calculate POH =  $4x10^8$  and115at J01D  $5x10^{-3}$ , POH = $7x10^8$ . In addition, we have to consider that when OH is consumed it leads to116HO2 formation, and the recycling of HO2 to OH will enhance the OH production beyond the primary production, especially in presence of moderate concentrations of (NO).

Based on the calculation above, we conclude that the observations of OH are expected andconsistent with the boundary conditions of the JPAC chamber.

Independent of that we doubt that coating would make a substantial difference to OH wall loss. Since OH is produced all the time by a local light source amid the chamber, there is always an OH gradient from lamp to wall. Indeed, there is likely a quite complex light and OH field in the chamber. However, as pointed out earlier, the chamber is continuously stirred with a mixing time of less than 2 minutes. This ensures all material with lifetimes >> 2 min in the chamber is homogenised and exposed to a uniform OH dose integrated over the entire OH field.

The intensity of the TUV lamp is relatively low; it is effective for OH production because  $O_3$  has a 129 cross section of 1.1x10-17 cm2 at 254 nm and a quantum yield of 1 for O1D production. Benzene has 130 absorptions in the spectral range of the TUV lamp, with a narrow maximum cross section at 253 nm of 131 8x10-18. However, this falls quickly off towards 254 nm where it is maximal with 1.4x10-18 at about 132 133 253.9 nm (Fally et al., 2009). Kamps et al., (1993) report guantum yields of close to zero when 134 shining benzene / inert gas mixtures with 254 nm low pressure HG lamps. This is in accordance with our observation of negligible benzene loss by photolysis when the TUV lamp was switched on. 135 Concentrations were monitored by PTR in dark and photolysis-only conditions (no oxidants). The 136 137 following sentences have been added to clarify these points in the text. 138

"The chamber is operated as a continuously stirred tank reactor with modifications as described in Mentel et al.
(2015). A fan made of Teflon provides typical mixing time of less than 2 minutes. The total flow in and out the chamber was 30 L min-1 resulting in an average residence time in the chamber of ~2900 seconds or 48 minutes."

**"The loss of benzene by photolysis was found to be negligible as measured by Q-PTR during experiments where only benzene was introduced into the chamber, in accordance with observations in the literature (Fally et al., 2009; Kamps et al., 1993)."**

The steady-state concentration of ozone is thus very important. Clearly, ozone does not react with 148 benzene, but once fragmentation occurs, there are plenty of double bonds for ozone to react with, 149 perhaps not as competitively as OH, but that only depends on the concentration of ozone being used. 150 This leads to my first major complaint. Please give the concentrations or concentration ranges being 151 used in the study. Better yet a table of the initial conditions: benzene, ozone, water (ppm), NO, NO2, 152 chamber residence time is essential. A table of initial conditions should not be relegated to a 153 supplementary section. It is ACP after all; there is no page limitation.

Table 1 of the main document contains the experimental details. These have been updated as
 requested. We double checked and recalculated the table input given in the ACPD version and
 removed erroneous numbers.

Onto the OH concentration in Table 1. I simply find it implausible that OH concentrations of 10(+8) 160 molecules can be formed in what is basically a smog chamber. (Obv-ously, high concentrations of OH 161 are generated in flowtubes operated on millisecond timescales. Not here.) 162 OH concentrations of up to 1E8 and higher can be produced in the JPAC chamber (e.g. Garmash et 163 al, 2020). This is due to the efficient O3 photolysis at 254nm as discussed in detail in the previous 164 165 comments. The average OH values in steady are calculated from the amount of benzene that was consumed, which is simply the difference of the benzene concentration in the inflow and the outflow. 166 The calculation of the OH concentration from benzene consumption was double checked by adding 167 168 1,8-cineole as a second OH tracer in an extra experiment. Moreover, the calculation of OH by the 169 consumption of hydrocarbons in general was verified in an earlier study by comparison to direct OH 170 measurements by LIF in the chamber (Broch, Thesis 2011). However, the value in the old Table 1 171 was a factor of 2 too large and has been revised. 172 173 There are just too many things for OH to do. As mentioned, walls and double-bonds are but two. This is all based on the p.5 equation for benzene loss and the low rate constant for irradiations over a 174 175 period of minutes results in high calculated OH levels. First, I would call Column 3, benzene loss 176 because that is what is measured. 177 178 Yes, the OH steady state concentration is determined by its source strength (J01D+H2O) as shown 179 above, and by all losses, including reactions with the reacted benzene and its oxidation products, as 180 well as losses to the wall. The chamber is permanently irradiated over the whole experiment period of several hours and OH is constantly produced. As mentioned above, benzene losses due to photolysis 181 182 at 254 nm are negligible. The steady state measurements were integrated over several minutes. We 183 have now clarified in the introduction that we are measuring the oxidation of benzene and its oxidation products, which includes unsaturated species and clarified the comments on wall losses. 184 185 186 Second, the absolute benzene loss is irrelevant. Thus, either the initial benzene concentration in 187 Table 1 caption or percent loss in the table should be given. In my opinion, the observed loss is some combination of reaction with OH and photolysis by 254 nm mercury radiation. This would thus lower 188 189 the calculated OH concentration. 190 191 The table of initial conditions has been updated. Experiments were conducted to verify the 192 concentration of OH by introduction of 1,8-cineole in addition to benzene. OH concentrations were 193 confirmed to be within 6 - 12% of the original values (Garmash et al., 2020). 194 The loss of benzene by photolysis at 254 nm is negligible, due to a near zero quantum yield. This was 195 further tested and verified in a dark experiment without oxidants, where benzene was introduced into 196 the chamber as in the actual experiments, and after reaching steady state benzene concentration, the 197 254nm lamp was switched on. In this experiment ozone had not been added to the chamber, so no 198 OH was produced; all benzene loss would be due to photolysis of the lamp. This experiment verified 199 the assumption based on literature that the photolysis of benzene at 254nm is negligible. 200 201 The following has been added to the text for clarification. 202 203 "Verification experiments were done by introducing 1.8-cineole into the chamber, which confirmed the 204 calculated OH concentrations to be within 6 - 12% (Garmash et al., 2020)." 205 206 However, if we take these concentrations seriously recognizing that the products are likely to be 207 considerably more reactive than the parent compound, it is easy to believe that 3-5 generations of products are likely to be present in the chamber with the CIMS measuring all of them, and perhaps 208 209 with increased oxidation, later generation products are being measured more sensitively than earlier 210 generation products. 211 212 We agree with the referee that benzene is interesting and challenging because its products react 213 faster with OH than benzene itself. We also agree that we obviously observe several generations of products and would claim that autoxidation (H-shifts in peroxy radicals) accelerates formation of 214 215 multiple generations in a short time (here 45 min). 216 Maybe it is important to remark here that we are fully aware that the CIMS sensitivity to the wide 217 218 range of compounds with different degrees of oxidation will intrinsically "never" be available. We make no attempt to quantify CIMS sensitivity as a function of product generation or claim that the CIMS
measures all oxidation products within the system. However, we observe many formula with
significant signal strength. We want to address the question of what can be learned from such CIMS
mass spectral patterns (measured with two different ionization schemes) for mechanistic
understanding. For example, we show that the iodide CIMS is not able to observe all products of
benzene oxidation suggested by the MCM, and show the difference in observed product distributions
compared to the nitrate ionisation, which is in agreement with other studies referenced in the
manuscript.

manuse 227

Of course, a major issue as noted in the manuscript is a lack of calibrations for products formed. This is unavoidable in these types of experiments. This is even worse when using a CIMS since the sensitivity of seemingly similar compounds can be one or more order-of-magnitude different. Therefore, to compare signal strength, as though implying, they are linearly related to concentration is a rapid path to mis-interpretation. I would recommend that graphics based on signal strength should be treated very carefully and worded conservatively.

As recognised here by the reviewer we state clearly that a lack of calibration prevents true 236 quantification. We aim at pattern analysis as mentioned in the previous comment. For these reasons 237 we make an honest attempt to present data in alternative yet still useful ways e.g. absolute numbers 238 of species, qualitative descriptions of those identified species, trends, as well as using analytical techniques that do not require calibrated data. Where signal is reported it is of course always within 239 240 the context of measurements by iodide CIMS and should be interpreted as such. As an aside, we 241 would claim that even if the sensitivity changes by an order of magnitude, the information about 242 hundreds of formula provides valuable information for mechanistic understanding of the system. It is 243 only a first step, though. Whilst it is true that due to the wide range of response factors a direct 244 equivalence cannot be made between iodide CIMS instrument response and concentration, it is 245 possible to infer broad trends for simple and well understood compounds. For example, iodide 246 ionisation has poor sensitivity to low O content species e.g. alcohols or ketones compared to 247 multifunctional and higher O containing functional groups such as carboxylic acids. In this manuscript 248 it is reported that ring retaining compounds containing 1 oxygen atom, i.e. phenol, consist of ~0.4 of 249 the total signal fraction in the low NOx experiment. Thus, if the sensitivity is low and the signal is high 250 then a broad conclusion can be made that this compound is present in 'large' amounts. Such 251 descriptions are not made in the manuscript because they are not properly quantifiable to any useful 252 degree, but nonetheless may be interpretable to those familiar with the measurement technique. This 253 of course becomes impossible when isomers and larger degrees of freedom are introduced.

Other iodide CIMS studies of aromatic systems have shown that cross calibrations and signal
distributions with other instruments (VOCUS) show that calibration factors for compounds with 1 to 6
oxygen atoms do not vary by orders of magnitude (Mehra et al., 2020). Whilst this would be an
assumption in this case, it is not an unreasonable inference that for broadly similar compounds,
sensitivities are not changing by orders of magnitude, but rather by a factor of <10.</li>

The comparison with MCM I find both tricky and the reverse of the experimental-deterministic model paradigm. First, given the complexity of MCM and the scarcity of quantitative data upon which later generation (beyond first) mechanisms are based I would describe the mechanisms at best as being tentative. It would be better to use the data in these experiments to provide some credence for later generation MCM product predictions, of course, not stating anything about predicted yields. Otherwise, it looks like "the blind leading the blind".

We agree with the reviewer that the mechanistic investigation is tentative and without calibrated data, predicted derived yields should not be attempted and they are not in this instance. We do not attempt 269 270 to investigate yields or branching ratios. We instead provide product distributions and molecular identification with some discussion to potential formation pathways expected from theoretical 271 272 mechanisms. We feel it is useful to compare the presence of these ions against those listed in MCM 273 as the MCM provides a comprehensive and often used standard of benzene oxidation chemistry. This 274 also provides insight into the usefulness of iodide-CIMS as an instrument for detecting benzene oxidation products and to compare with other CIMS studies of aromatic systems such as those 275 276 referenced in the manuscript. At the same time the CIMS data indicate where the MCM concepts may 277 fall short. 278

Something needs to be added regarding the possible role of NO3 radicals in the system, especially
given the number of oxidized nitrogen compounds being detected. Given what I expect to be a high
ozone concentration (ppm), its reaction with NO2 is probably occurring to some non-negligible
degree.

Steady state ozone concentrations are measured at a maximum of 60 ppb. This has now been added
to table 1. NO3 cannot be build up under the given conditions, since loss of NO3 is rapid due to N2O5
formation and subsequent loss to surfaces due to the 60% humidity. This renders the impact of NO3
radicals negligible.

I see no evidence provided that autooxidation plays an important role in the system. Perhaps, it does,
but not from the data presented. Of course, it may be that the authors have considered and
addressed the issues above and similar issues. However, these issues are critical to understanding
how the data are interpreted and must be included in the paper, if the study is to have any credibility.

The reviewer is correct, separating the role of multiple oxidations by OH and autoxidation is not easy in benzene oxidation. This is extensively addressed in our paper by Garmash et al., 2020. Here we do not attempt to describe the relative importance of autoxidation compared to other mechanistic pathways. We describe product distributions and discuss potential autoxidation products and their distribution within the system, under different NOx conditions. We don't see a need to repeat material published by Garmash et al., but we will give more reference to it.

- Finally, I believe it would be valuable to show how the findings from this paper fit into the
  understanding of benzene oxidation.
- 302 303

As discussed previously it is impossible to discuss the finding of this paper in a quantitative way, which leaves qualitative descriptions. Qualitative presentations of product distributions and ion identifications are made and a discussion of the potential origins of those observed ions in terms of widely accepted chemical mechanisms is also made.

**Thus, it is my opinion that this manuscript needs substantial work before it is published.**

311 We addressed all points by the referee which we think were often based on misunderstandings 312 because of our incomplete description of our setup and boundary conditions. We think that we have 313 addressed these issues in this response. We give more detailed information on the steady state concentrations of major reactants as requested. We believe that with this input it will become clear 314 315 that the study is within the given limits of quantification. From these points of view we do not believe 316 that this manuscript requires more work, beyond addressing the reviewers' comments and added clarifications to the manuscript. Especially if one considers that much of the requested information 317 was already published previously in refereed journals, like the papers by Mentel et al., ACP (2009, 318 2015) (regarding the chamber performance) or Garmash et al., ACP (2020) regarding benzene 319 320 oxidation and autoxidation.

Regardless of the previously published information, we have tried to make the manuscript more
 coherent by including more information and reorganisation. Especially we have reorganised the
 structure of section 3 and we included more illustrative text to better qualify the aims of the study and
 to improve clarity and to better quide the reader.

- 325326 A few line-by-line comments.
- 327

Line 77. Johnson et al. 2005 is a rather poor reference. Dicarbonyl aldehydes were shown to be formed in aromatic systems by the mid-1980's (e.g., Dumdei and O'Brien, Science, 1983; Shepson et al. JPC, 1984; Dumdei et al, EST, 1988.)

- 331
- The reference has been updated.

Line 132. I am not sure what this sentence is saying. Mass closure, if that is ever possible, will require a lot more than complementary instruments. In fact, as these experiments are conducted, each product generation probably gets you further and further away from "mass closure". I would reword the sentence in a more realistic fashion and perhaps with a bit less jargon.

We believe the use of the word mass here is confusing and unhelpful. The word mass is removed andthe sentence is updated to be clearer:

"However, to observe carbon in a diverse range of forms, e.g. different oxidation states and of different functionalities, multiple ionisation schemes are required as they are sensitivities towards different OVOCs (e.g.
Isaacman-VanWertz *et al.*, 2017; Riva *et al.*, 2019). These methods, combined with other measurement
techniques, have been demonstrated to enable the complete observation of all the reacted carbon within a system
(carbon closure) (Isaacman-Vanwertz et al., 2018)."

Line 148-49, Table 1. Is it true that the reacted benzene concentrations were identical to three
significant figures for the experiments shown? The actual measured data would be more helpful
together with the percent benzene loss.

The table of initial conditions has been updated and revised.

Line 153. The sentence beginning in this line is a "red herring". Otherwise, reactions of benzene with ozone in the chamber would lead to product distributions being a complete mess. If benzene and ozone were introduced together, there would probably be less of a mixing issue.

The words "in order to prevent reactions occurring inline" have been removed.

Line 204. How about calibrations for the instrument sensitivity for the inorganic nitro-geneous compounds? Were any undertaken?

We try to monitor one easily calibrated organic compound (here formic acid) during measurements to assess sensitivity changes throughout the experiment as an indicator of instrument performance. Although this information is not used explicitly to calculate any concentrations, it may be useful for others to assess the performance of our instrument during these experiments and so is presented here. We do not deem the quantification of other inorganic nitrogen compounds by CIMS necessary to the work presented here.

Line 246. I am not sure what the term "time series" means in the context of these experiments. They
were all done under steady-state conditions.

The reviewer states "Thus, three or four residence times are typically needed to achieve steady state". Thus, they acknowledge the dynamic nature of the system before steady state is achieved. We 374 375 take advantage of this and use the information we can gather from before steady state to aid 376 understanding of molecular identification. This is discussed in terms of product type and oxidation. 377 Beyond photostationary steady state the reaction produces semi-volatile and condensable material that deposit and equilibrate with sample line walls. This partitioning between condensed and gas 378 phase is detected here and discussed appropriately within the context of similar literature. We also 379 show the usefulness of clustering time series, similar to other studies referenced in the manuscript, 380 within the context of an experiment with well-defined and different conditions, in this case different 381 382 NOx fractions. We have added the following sentence clarifying the analysis: 383

"We use hierarchical clustering analysis (HCA) of the time series' from the iodide ToF-CIMS to explore the
effects of different NOx conditions on the instrument response to different organic nitrogen containing
products."

Line 275. This would be an excellent place to include a table for the initial conditions. This would also allow us to see how many experiments the data is based on. One of my main questions is how many different residence times were tested. For example, it would have been helpful to have one experiment  $at_\tau = 1450$  s and another  $at_\tau = 5800$ s. (Note: I am not asking for any additional experiments if only one residence time was tested.)

Updated table 1 contains the details of initial conditions including number of experiments. Only one
residence time was tested. Testing different residence times was beyond the scope of this study.

Line 280. Again, I am not sure what "time-series behavior" means when conducting experiments in a steady-state reactor.

This point is addressed in an earlier response: we monitor also the transient periods, i.e. the rise time 401 of products, although most data was analysed for steady state conditions 402 403 Line 283. The other studies mentioned were probably performed with the experiment(s) being conducted in a batch mode. However, for a CSTR used in reactive systems, abroad spike occurs at 404 405 the beginning solely from the dynamics of a reacting system reaching steady state. Thus, three or four residence times are typically needed to achieve steady state. Moreover, I suspect the system is not 406 fully homogeneous and certainly not at the beginning of the irradiation. OH is certainly a problem -407 408 high concentrations near the lamp and low concentrations near the wall. There is also the question as 409 to whether the radiation is optically thin. I believe there are many factors leading to this "spiking" with 410 gas-aerosol partitioning probably being a minor one. 411 412 This spiking behaviour is discussed in Garmash et al., 2020. The manuscript also states 413 414 "The early spiking also describes the dynamic nature of oxidation concentrations before the steady state is 415 reached." 416 417 This is in line with the reviewers comment here, and we have re-written the sentence to improve its 418 clarity. 419 420 "This early transient maximum has been observed in other studies investigating VOC oxidation using the nitrate 421 ToF-CIMS (e.g. Ehn et al. 2014) and is thought to be a consequence of a mixture of phenomena that could 422 include a latent aerosol sink shifting the equilibrium species from the gas to the aerosol phase as well as the 423 dynamic nature of oxidant concentrations (mainly adaptation of  $O_3$ ) before the steady state is reached and 424 mixing is totally homogenous." 425 426 Line 381. I find this section a bit too speculative and should be written more conservatively. I again repeat my comments regarding MCM and the CIMS data. Certainly, the CIMS data has no 427 428 quantitative significance, since the sensitivities are typically all-over-the-map. This section would be a 429 good place to consider what the uncertainties are present in these experiments. 430 431 The mechanistic investigation is described in the manuscript as theoretical and within the restrictions 432 of the MCM mechanism with the additional autoxidation mechanism applied: the framework of MCM + autoxidation is clearly defined. We believe this clearly indicates the constraints placed on the 433 investigation in this section. At no point do we attempt to quantify anything other than the number of 434 435 ions observed and their formulae and discuss them in terms of the mechanism described. We 436 improve the clarity of this section to ensure the reader is aware of the limitations of the investigation. 437 438 "This was repeated for phenol and catechol precursors as products of subsequent OH attack (Garmash et al., 439 2020). To reduce complexity, this subsequent OH attack is only considered at the beginning of the mechanism. 440 This provides a total of 53 individual oxidation products with 21 unique formula." 441 442 Line 421. A consideration of the photolysis of the nitrogenated products at 254 nm might be 443 considered in this section as well. 444 445 Unfortunately not much data exist on the absorption cross sections of CHON compounds at 254 nm 446 and almost none of quantum efficiencies in this wavelength range (Keller-Rudek et al., 2013). Robers 447 and Fajer (1989) and Talukdar et al. (1997) report cross section of several organic nitrates of a few times 10-20s-10-19. For methylnitrate quantum yields for NO2 formation are close to one at 248 nm. 448 449 Therefore, unlike with benzene, we cannot exclude minor contributions of photolysis of organic 450 nitrates at 254 nm to the product spectrum. However, many aromatic N compounds are known to 451 absorb at higher wavelengths (300 - 400 nm) (Laskin et al., 2015) and so we expect the highest 452 absorption cross section of aromatic N containing compounds to be shifted to these higher 453 wavelengths and not to be impacted by the weak and coherent 254 nm TUV emission so much. 454 455 Line 470. The first sentence of the conclusion is totally unsupported. The sentence reads as though 456 NOx is decoupled from OH. Experiments having steady-state OH concentrations near 10(+8) molec/cc have at best marginal relevance even with atmospherically plausible levels of NOx being 457 458 tested. I would be far more circumspect in making atmospherically relevant statements from this dataset. At minimum, additional qualifying statements need to be provided in this paragraph. I wouldhighly recommend simply leaving out the first paragraph.

We agree and the qualification "atmospheric relevant" may be overstated in this context. We will remove it from the sentence. However, OH concentrations of up to 107 can be observed in the atmosphere. Whilst the OH concentrations in these experiments are 5-10 times higher than typical atmospheric concentrations, integrating the OH concentration over the residence time gives a total dose of OH which can then be equated to an equivalent exposure time at an atmospherically relevant concentration. This has been clarified with the additional statements and added into the initial conditions table:

"Integrating the OH concentrations over the residence time gives an equivalent OH dosage that may be compared
to atmospheric levels. Here OH equivalent doses are equivalent to approximately 5 days of oxidation at a relevant
atmospheric OH concentration (106 molecules cm-3)."

Line 486ff. From this point on, I think the data is being overinterpreted.

This includes most of the conclusion, including purely descriptive sentences that summarise the same
points communicated in the relevant sections of the text. Without any specific comments, this is
difficult to address. We underline more clearly the qualitative character and the limit that the sensitivity
can also change the relative importance. The following line has been added to improve clarification of
interpretation

**481 "It is stressed that as these values are reported as uncalibrated instrument responses, they cannot be used to482 directly assess chemical pathways."**

484

**485 Reviewer 2**

However, the major issue is that the discussions are fragmented. Many interesting observations are presented, but it is a bit blurred how such detailed measurements of nearly 200 ions improve our understanding on the fundamental oxidation mechanism of benzene, besides leaving the impressions that the benzene oxidation generates hundreds of products and the oxidation products are different between low- and high-NOx conditions. The manuscript can be largely improved if the detailed discussions can be synthesized in a coherent fashion. Overall, I recommend publication after major revision

We agree with the reviewer that the structure of the paper can be made more coherent. We have
reorganised the structure and included more illustrative text to better qualify the aims of the study and
to improve clarity and to better guide the reader.

- 497498 In this study, the oxidation of benzene and its oxidation products by OH under high and low NOx conditions are
  - 499 investigated in the Jülich plant atmosphere chamber (JPAC) (Mentel et al., 2009, 2015) with two time of flight
  - 500 chemical ionisation mass spectrometers (ToF-CIMS) using the iodide and nitrate ionisation schemes. The
  - 501 ionisation schemes are compared to assess the similarities in detected oxidation products in terms of molecular
  - 502 identification and bulk properties. Mechanistic investigations to assess the ability of the iodide CIMS to detect
  - 503 species that currently aren't accounted for i.e. HOM and those described in the MCM are investigated.
  - 504 Additionally high and low  $NO_x$  conditions are probed. As sensitivities for the iodide CIMS are lacking, an
  - 505 exploration of product descriptions is made using bulk analysis, as well as broad groupings defined by bulk
  - 506 properties. Finally, the application of hierarchical clustering analysis (HCA) is used to provide the basis of a

- 507 methodology to assess the time series behaviour of oxidation products that can further aid molecular identification508 e.g. by inferring potential functionality.
- 509 Moved mechanistic investigation section to 3.2 and added the text
- 510 "Nitrate CIMS is routinely used to detect HOM, but iodide CIMS is also able to detect high mass compounds
- (Mohr et al., 2019). To test the extent to which the iodide CIMS is able to detect HOM that could potentiallyoriginate from autoxidative processes, a number of theoretically suggested formulae based on the autoxidation
- 513 mechanism with propagation and termination steps were devised and then searched for within the spectra."
- 514 "HOM and autoxidation are currently not a common inclusion in chemical box models, although some
- mechanisms are available (e.g. Weber *et al.*, 2020). Due to its usage and optimisation for HOM detection, nitrate
- 516 CIMS does not typically provide insight into traditional oxidation chemistry, as its sensitivity to low molecular
- 517 weight species is poor. In order to assess the traditional oxidation chemistry of this system, we describe which
- 518 oxidation products from the MCM (Jenkin *et al.*, 2003; Bloss *et al.*, 2005) are detectable by iodide CIMS."
- 519 Moved MCM products to section to 3.2.1 and added the text
- 520 "Without calibration, much of the oxidation product descriptions must be either; tentative, when making direct
- 521 identifications; or broad, and describe bulk organic properties. In the absence of direct calibration, we describe
- 522 the bulk properties of iodide CIMS detected CHO and CHON compounds and group them according to chemical
- 523 composition criteria in order to present an overall depiction of all detectable oxidation products."
- 524 Moved Iodide CIMS detected CHO and CHON under different NOx conditions to section 3.3
- 525 Moved Ring retention and ring breaking products to section to 3.3.1 and added the text
- 526 "Descriptions of bulk properties and groupings into large families are a useful method to assess broad trends
- 527 within the system. However, it may be possible to produce less granularity in identified species by looking at
- 528 specific atomic composition and attempting to infer potential functionality."
- 529 Added text to the conclusion.
- 530 "This methodology used in conjunction with an internal calibration procedure such as voltage scanning (Lopez-
- 531 Hilfiker et al., 2015) could then be used to quantitatively describe shifts in CHON composition as a function of
- 532 NO/NOx and further investigate the relative importance of species isomers and their formation pathways to a mass
- 533 spectrometric signal."
- 534 Comments

- 1. Many products more than 6 carbons (like C8, C9) have been detected. Please discuss the potential formation mechanisms of these compounds.
- 538 The following sentence has been added.

"Detection of molecules with greater than C6 that are not C12 dimers are likely formed through peroxy radical cross reactions."

Some comparisons between measurement and MCM are not conducted in a meaningful way.
 For example, in Page 10 Line 351, it is claimed that compounds of formulae that match
 species in MCM compare 7.3% and 6.4% of the low and high NOx experiments, respectively.

There are two issues in this comparison. First, the values depend on the extent of oxidation. 547 For example, it is well-studied that the phenol yield in benzene oxidation~50%. MCM 548 compounds should least comprise 50% of detected compounds when the secondary chemistry is negligible. Second, as the response factors in I-CIMS vary by orders of 549 magnitude, the raw signal in Hz cannot represent the true product distribution. 550 551 552 We agree that the inclusion of precise percentages here, whilst correct, over emphasise the relevance 553 of signal and can be misinterpreted as a true quantification, which is not the intention. We leave the 554 percentages in the results text because they directly describe the figure (9) which has other 555 interesting features as described in the text. However, we reduce the precision of the percentages 556 557 and present them more approximately and state clearly that these relate to ion signal and not 558 concentration. The percentages are entirely removed from the conclusion and described more 559 qualitatively to prevent misinterpretation. 560 561 Other iodide CIMS studies of aromatic systems have shown that cross calibrations and signal distributions with other instruments (VOCUS) show that calibration factors for compounds with 1 to 6 562 563 oxygen atoms do not vary by orders of magnitude (Mehra et al., 2020). Whilst this would be an 564 assumption in this case, it is not an unreasonable inference that for broadly similar compounds, sensitivities are not changing by orders of magnitude, rather by a factor of <10. See also the response 565 566 to reviewer 1 on the broader inferences that can be made when interpreting CIMS signal, i.e. how high signal for low sensitivity species such as C6H6O suggests high concentrations. 567 568 569 Similarly, because of the two issues mentioned above (i.e., uncertainties in instrument 570 sensitivity and extent of oxidation), it is unclear how meaningful the reported distribution of 571 products is. The discussion on Page 10 Line 359 is one example. 572 The oxidation is now defined in terms of OH dosage. The distributions are meaningful in terms of the 573 574 measurement techniques, which are contrasted and discussed. The following clarification on how these numbers and figures can be interpreted has been added to the text. 575 576 577 "It is stressed that as these values are reported as uncalibrated instrument responses they cannot be used to 578 directly assess chemical pathways. However, they allow for the comparison of product distributions between 579 high and low NOx conditions in terms of sensitivity to iodide CIMS measurements (Fig 10 and Fig 11)." 580 581 3. Page 11 Line 375. How are these two values calculated? 582 These are calculated from the numbers in the paragraph above; of the 85 oxidation products listed in 583 the MCM, 19 and 26 formulae are detected in each NOx case. The line has been removed and the 584 information incorporated into the previous paragraph. 585 586 "The iodide CIMS is able to observe 25 % and 30 % of the listed MCM compounds in the low and high NO. 587 conditions respectively (Fig 8)." 588 589 590 "The iodide ToF-CIMS detects 19 ( $\sim$ 25 %) and 26 ( $\sim$ 30 %) of these oxidation products under low and high NOx 591 conditions (Fig 8)." 592 593 4. Page 11Line 396-397. This statement on the potential formation mechanism of C6H8O6, etcis too strong. The HOMs formation mechanism from benzene oxidation is unclear. For example, 594 595 Garmash et al. 1 showed that the HOMs yield is higher in benzene oxidation than phenol oxidation. It lacks support to state that C6H8O6, etc can only be formed from phenol or 596 597 catechol. 598 599 The context within which the statement in this section is made is based upon the current mechanism found within the MCM and the defined autoxidation method outlined in the paragraph above. The 600 following sentences have been added to clarify this: 601 602 603 "This was repeated for phenol and catechol precursors as products of subsequent OH attack (Garmash et al., 604 2020). To reduce complexity, this subsequent OH attack is only considered at the beginning of the mechanism." 605

"All the suggestions of potential mechanistic routes to formation are speculative and set against the mechanistic 607 paradigm laid out here.' 608 609 We agree this statement lacks support in terms of precursor and have revised the sentence. 610 611 "Here,  $C_6H_8O_6$ ,  $C_6H_8O_7$ ,  $C_6H_8O_8$  are potentially formed through the autoxidation mechanism" 612 613 5. Table 1. Please include the initial concentration of benzene. I estimate that roughly 20-40% of initial benzene is oxidized in the experiments. Because benzene oxidation products are much 614 more reactive than benzene, many detected products are likely from multi-generation 615 chemistry. This should be clearly mentioned in the manuscript. 616 617 618 Table 1 has been updated to include the initial concentrations. The following sentence in the introduction is 619 updated to include benzene oxidation products. 620 "In this study, the oxidation of benzene and its oxidation products by OH under high and low NOx conditions 621 622 are investigated in the Jülich plant atmosphere chamber (JPAC) with two time of flight chemical ionisation mass 623 spectrometers (ToF-CIMS) using the iodide and nitrate ionisation schemes." 624 625 References Bloss, C., Wagner, V., Jenkin, M. E., Volkamer, R., Bloss, W. J., Lee, J. D., Heard, D. E., Wirtz, K., Martin-626 Reviejo, M., Rea, G., Wenger, J. C. and Pilling, M. J.: Development of a detailed chemical mechanism 627 (MCMv3.1) for the atmospheric oxidation of aromatic hydrocarbons, Atmos. Chem. Phys., 5(3), 641–664, 628 629 doi:10.5194/acp-5-641-2005, 2005. 630 Ehn, M., Thornton, J. A., Kleist, E., Sipilä, M., Junninen, H., Pullinen, I., Springer, M., Rubach, F., Tillmann, 631 R., Lee, B., Lopez-Hilfiker, F., Andres, S., Acir, I.-H., Rissanen, M., Jokinen, T., Schobesberger, S., Kangasluoma, J., Kontkanen, J., Nieminen, T., Kurtén, T., Nielsen, L. B., Jørgensen, S., Kjaergaard, H. G., 632 633 Canagaratna, M., Maso, M. D., Berndt, T., Petäjä, T., Wahner, A., Kerminen, V.-M., Kulmala, M., Worsnop, D. 634 R., Wildt, J. and Mentel, T. F.: A large source of low-volatility secondary organic aerosol, Nature, 506(7489), 635 476-479, doi:10.1038/nature13032, 2014. 636 Fally, S., Carleer, M. and Vandaele, A. C.: UV Fourier transform absorption cross sections of benzene, toluene, 637 meta-, ortho-, and para-xylene, J. Quant. Spectrosc. Radiat. Transf., 110(9-10), 766-782, 638 doi:10.1016/j.jqsrt.2008.11.014, 2009. 639 Garmash, O., Rissanen, M. P., Pullinen, I., Schmitt, S., Kausiala, O., Tillmann, R., Zhao, D., Percival, C., 640 Bannan, T. J., Priestley, M., Hallquist, Å. M., Kleist, E., Kiendler-Scharr, A., Hallquist, M., Berndt, T., 641 McFiggans, G., Wildt, J., Mentel, T. F. and Ehn, M.: Multi-generation OH oxidation as a source for highly 642 oxygenated organic molecules from aromatics, Atmos. Chem. Phys., 20(1), 515-537, doi:10.5194/acp-20-515-643 2020, 2020. 644 Isaacman-Vanwertz, G., Massoli, P., O'Brien, R., Lim, C., Franklin, J. P., Moss, J. A., Hunter, J. F., Nowak, J. B., Canagaratna, M. R., Misztal, P. K., Arata, C., Roscioli, J. R., Herndon, S. T., Onasch, T. B., Lambe, A. T., 645 Jayne, J. T., Su, L., Knopf, D. A., Goldstein, A. H., Worsnop, D. R. and Kroll, J. H.: Chemical evolution of 646 647 atmospheric organic carbon over multiple generations of oxidation, Nat. Chem., 10(4), 462–468, 648 doi:10.1038/s41557-018-0002-2, 2018. 649 Isaacman-VanWertz, G., Massoli, P., O'Brien, R. E., Nowak, J. B., Canagaratna, M. R., Jayne, J. T., Worsnop, 650 D. R., Su, L., Knopf, D. A., Misztal, P. K., Arata, C., Goldstein, A. H. and Kroll, J. H.: Using advanced mass spectrometry techniques to fully characterize atmospheric organic carbon: current capabilities and remaining 651 gaps, Faraday Discuss., 200, 579-598, doi:10.1039/C7FD00021A, 2017. 652 653 Jenkin, M. E., Saunders, S. M., Wagner, V. and Pilling, M. J.: Protocol for the development of the Master 654 Chemical Mechanism, MCM v3 (Part B): Tropospheric degradation of aromatic volatile organic compounds, 655 Atmos. Chem. Phys., 3(1), 181–193, doi:10.5194/acp-3-181-2003, 2003. Kamps, R., Müller, H., Schmitt, M., Sommer, S., Wang, Z. and Kleinermanns, K.: Photooxidation of exhaust 656 pollutants. I. Degradation efficiencies, quantum yields and products of benzene photooxidation, Chemosphere, 657 27(11), 2127-2142, doi:10.1016/0045-6535(93)90125-O, 1993. 658 659 Keller-Rudek, H., Moortgat, G. K., Sander, R. and Sörensen, R.: The MPI-Mainz UV/VIS spectral atlas of 660 gaseous molecules of atmospheric interest, Earth Syst. Sci. Data, 5(2), 365–373, doi:10.5194/essd-5-365-2013, 661 2013. Laskin, A., Laskin, J. and Nizkorodov, S. A.: Chemistry of Atmospheric Brown Carbon, 662 doi:10.1021/cr5006167, 2015. 663 Lopez-Hilfiker, F. D., Iyer, S., Mohr, C., Lee, B. H., D'Ambro, E. L., Kurtén, T. and Thornton, J. A.: 664 665 Constraining the sensitivity of iodide adduct chemical ionization mass spectrometry to multifunctional organic

- molecules using the collision limit and thermodynamic stability of iodide ion adducts, Atmos. Meas. Tech.
  Discuss., 8(10), 10875–10896, doi:10.5194/amtd-8-10875-2015, 2015.
- 668 Mehra, A., Wang, Y., Krechmer, J., Lambe, A., Majluf, F., Morris, M., Priestley, M., Bannan, T., Bryant, D.,
- 669 Pereira, K., Hamilton, J., Rickard, A., Newland, M., Stark, H., Croteau, P., Jayne, J., Worsnop, D., Canagaratna,
- 670 M., Wang, L. and Coe, H.: Evaluation of the Chemical Composition of Gas and Particle Phase Products of
- 671 Aromatic Oxidation, Atmos. Chem. Phys., (March), 1–24, doi:10.5194/acp-2020-161, 2020.
- 672 Mentel, T. F., Wildt, J., Kiendler-Scharr, A., Kleist, E., Tillmann, R., Dal Maso, M., Fisseha, R., Hohaus, T.,
- 673 Spahn, H., Uerlings, R., Wegener, R., Griffiths, P. T., Dinar, E., Rudich, Y. and Wahner, A.: Photochemical
- production of aerosols from real plant emissions, Atmos. Chem. Phys., 9(13), 4387–4406, doi:10.5194/acp-94387-2009, 2009.
- 676 Mentel, T. F., Springer, M., Ehn, M., Kleist, E., Pullinen, I., Kurtén, T., Rissanen, M., Wahner, A. and Wildt, J.:
- 677 Formation of highly oxidized multifunctional compounds: autoxidation of peroxy radicals formed in the
- ozonolysis of alkenes deduced from structure–product relationships, Atmos. Chem. Phys., 15(12), 6745–6765,
- 679 doi:10.5194/acp-15-6745-2015, 2015.
- 680 Mohr, C., Thornton, J. A., Heitto, A., Lopez-Hilfiker, F. D., Lutz, A., Riipinen, I., Hong, J., Donahue, N. M.,
- 681 Hallquist, M., Petäjä, T., Kulmala, M. and Yli-Juuti, T.: Molecular identification of organic vapors driving
- 682 atmospheric nanoparticle growth, Nat. Commun., 10(1), 1–7, doi:10.1038/s41467-019-12473-2, 2019.
- 683 Riva, M., Rantala, P., Krechmer, E. J., Peräkylä, O., Zhang, Y., Heikkinen, L., Garmash, O., Yan, C., Kulmala,
- 684 M., Worsnop, D. and Ehn, M.: Evaluating the performance of five different chemical ionization techniques for
- detecting gaseous oxygenated organic species, Atmos. Meas. Tech., 12(4), 2403–2421, doi:10.5194/amt-12 2403-2019, 2019.
- 687

---

## Author Response (AR2)

We thank the reviewer for their comments and for highlighting the paper by Xu et al., 2020. The following points have been added to the manuscript:

The dominant fate of this alkoxy radical is ring breaking which can further form a variety of SOA relevant compounds such as HOMs, glyoxal, and various $C_4$ and $C_5$ compounds (Xu *et al.*, 2020).

$C_5$ compounds are shown to form from the ring opening of the di-oxygen bridge alkoxy radical (Xu *et al.*, 2020) and could subsequently be involved in dimer formation of $C_9$ and $C_{10}$ compounds.

It should be noted that these species are identified in Xu *et al.* (2020) as products of an epoxidation channel of the alkoxy phenol radical.

Contrary to the comment, C6H7NO5 (BCE nitrate) is not observed in this study although C6H7NO4 (BCP nitrate) and C6H7NO6 (TCEE) are. These latter two compounds could be formed through the epoxy channel, however in the cluster analysis of this paper they are assigned to cluster 3 which does not form part of the discussion on how NO/NO$_x$ fractions impact RNO$_x$ product distributions. The following passage is added to discuss this point:

Additionally, the presence of a pathway to form N containing ring retaining epoxides from the alkoxy phenyl radical recently described by Xu *et al.* (2020) further demonstrates the difficulty of structural assignments and accounting for the position of oxygen atoms. $C_6H_7NO_4$ and $C_6H_7NO_6$ are two formulae that could represent ring retaining nitrogen containing epoxides however as these two formulae are found in cluster 3 they do not impact the discussion on the effect of NO/NO$_x$ ratios on clusters 1 and 2 here.